

# Biogeophysical impacts of forestation in Europe: First results from the LUCAS Regional Climate Model intercomparison

Edouard L. Davin[1], Diana Rechid[2], Marcus Breil[3], Rita M. Cardoso[4], Erika Coppola[5], Peter Hoffmann[2,6], Lisa L. Jach[7], Eleni Katragkou[8], Nathalie de Noblet-Ducoudré[9], Kai Radtke[10], Mario Raffa[11], Pedro M.M. Soares[4], Giannis Sofiadis[8], Susanna Strada[5], Gustav Strandberg[12], Merja H. Tölle[13], Kirsten Warrach-Sagi[7], Volker Wulfmeyer[7]

[1]ETH Zurich, Zurich, Switzerland
[2]Climate Service Center Germany (GERICS), Helmholtz-Zentrum Geesthacht, Hamburg, Germany
[3]Institute of Meteorology and Climate Research, Karlsruhe Institute of Technology, Karlsruhe, Germany
[4]Instituto Dom Luiz (IDL), Faculdade de Ciências, Universidade de Lisboa, 1749-016 Lisboa, Portugal
[5]International Center for Theoretical Physics (ICTP), Earth System Physics Section, Trieste, Italy
[6]Helmholtz-Institut Climate Service Science (HICSS), Hamburg, Germany
[7]Institute of Physics and Meteorology, University of Hohenheim, Stuttgart, Germany
[8]Department of Meteorology and Climatology, School of Geology, Aristotle University of Thessaloniki, Thessaloniki, Greece.
[9]Laboratoire des Sciences du Climat et de l'Environnement; UMR CEA-CNRS-UVSQ, Université Paris-Saclay; Orme des Merisiers, bât 714; 91191 Gif-sur-Yvette cédex; France
[10]Chair of Environmental meteorology, Brandenburg University of Technology, Cottbus-Senftenberg, Germany
[11]REgional Models and geo-Hydrological Impacts, Centro Euro-Mediterraneo sui Cambiamenti Climatici, Italy
[12]Swedish Meteorological and Hydrological Institute and Bolin Centre for Climate Research, Norrköping, Sweden
[13]Department of Geography, Climatology, Climate Dynamics and Climate Change, Justus-Liebig University Giessen, Giessen, Germany

Correspondence to: Edouard L. Davin (edouard.davin@env.ethz.ch)

**Abstract.** The Land Use and Climate Across Scales Flagship Pilot Study (LUCAS FPS) is a coordinated community effort to improve the integration of Land Use Change (LUC) in Regional Climate Models (RCMs) and to quantify the biogeophysical effects of LUC on local to regional climate in Europe. In the first phase of LUCAS, nine RCMs are used to explore the biogeophysical impacts of re-/afforestation over Europe. Namely, two idealized experiments representing respectively a non-forested and a maximally forested Europe are compared in order to quantify spatial and temporal variations in the regional climate sensitivity to forestation. We find some robust features in the simulated response to forestation. In particular, all models indicate a year-round decrease in surface albedo, which is most pronounced in winter and spring at high latitudes. This results in a winter warming effect, which is relatively robust across models. However, there are also a number of strongly diverging responses. For instance, there is no agreement on the sign of temperature changes in summer with some RCMs predicting a widespread cooling from forestation, a widespread warming, or a mixed response. A large part of the inter-model spread is attributed to the representation of land processes. In particular, differences in the partitioning of sensible and latent heat are identified as a key source of uncertainty. In contrast, for precipitation, the representation of atmospheric processes dictates more directly the simulated response. In conclusion, the multi-model approach we use here has the potential to deliver more



robust and reliable information to stakeholders involved in land use planning, as compared to results based on single models. However, given the contradictory responses identified, our results also show that there are still fundamental uncertainties that
need to be tackled to better anticipate the possible intended or unintended consequences of LUC on regional climates.

## 1 Introduction

Land Use Change (LUC) affects climate through biogeophysical processes influencing surface albedo, evapotranspiration and surface roughness (Bonan 2008; Davin and de Noblet-Ducoudre 2010). The quantification of these effects is still subject to particularly large uncertainties, but there is growing evidence that LUC is an important driver of climate change at local to
regional scales. For instance, the Land-Use and Climate, IDentification of robust impacts (LUCID) model intercomparison indicated that while LUC likely had a modest biogeophysical impact on global temperature since the pre-industrial era, it may have affected temperature in similar proportion as greenhouse gas forcing in some regions (de Noblet-Ducoudre et al. 2012). Results from the Coupled Model Intercomparison Project Phase 5 (CMIP5) confirmed the importance of LUC for regional climate trends and for temperature extremes (Lejeune et al. 2017, 2018; Kumar et al. 2013).


In this light, it is particularly important to represent LUC forcings in regional climate simulations. Yet, LUC forcings were not included in previous RCM intercomparisons (Christensen and Christensen 2007; Jacob et al. 2014; Mearns et al. 2012; Solman et al. 2013), which are the basis for numerous regional climate change assessments providing information for impact studies and the design of adaptation plans (Gutowski Jr. et al. 2016). RCMs have been applied individually to explore different aspects
of land use impacts on regional climates (Gálos et al. 2013; Davin et al. 2014; Lejeune et al. 2015; Wulfmeyer et al. 2014; Tölle et al. 2018), but the robustness of such results is difficult to assess due to their reliance on single RCMs and to the lack of a common protocol. There is therefore a need for a coordinated effort to better integrate LUC effects in RCM projections. The Land Use and Climate Across Scales (LUCAS) initiative (https://www.hzg.de/ms/cordex-fps-lucas/) has been designed with this goal in mind. LUCAS is endorsed as a Flagship Pilot Study (FPS) by the World Climate Research Program-
Coordinated Regional Climate Downscaling Experiment (WCRP-CORDEX) and was initiated by the European branch of CORDEX (EURO-CORDEX) (Rechid et al. 2017). The objectives of the LUCAS FPS are to promote the inclusion of the missing LUC forcing in RCM multi-model experiments and to identify the associated impacts with a focus on regional to local scales and considering time scales from extreme events to seasonal and multi-decadal trends and variability. LUCAS is designed in successive phases that will go from idealized to realistic high-resolution scenarios and intend to cover both land
cover changes and land management impacts.

In the first phase of LUCAS, which is the focus of this study, idealized experiments over Europe are performed in order to benchmark the RCMs sensitivity to extreme LUC. Two experiments (FOREST and GRASS) are performed using a set of nine RCMs. The FOREST experiment represents a maximally forested Europe, while in the GRASS experiment trees are replaced





by grassland. Comparing FOREST to GRASS therefore indicates the theoretical potential of a maximum forestation (encompassing both reforestation and afforestation) scenario over Europe. Given that forestation is one of the most prominent land-based mitigation strategies put forward in scenarios compatible with the Paris Agreement goals (Grassi et al. 2017; Harper et al. 2018; Griscom et al. 2017), it is therefore essential to understand its full consequences beyond $CO_2$ mitigation. These experiments are not meant to represent realistic scenarios but they enable a systematic assessment and mapping of the

biogeophysical impact of forestation across regions and seasons. Experiments of this type have already been performed using single regional or global climate models (Claussen et al. 2001; Davin and de Noblet-Ducoudre 2010; Cherubini et al. 2018; Strandberg et al. 2018), but here they are performed for the first time using a multi-model ensemble approach, thus providing an unprecedented opportunity to assess uncertainties in the climate response to vegetation perturbations. In the following, we focus on the analysis of the surface energy balance and temperature response at the seasonal time scale, while future studies

within LUCAS will explore further aspects (e.g. sub-daily time scale and extreme events, land-atmosphere coupling, etc). We aim to quantify the potential effect of forestation over Europe, identify robust model responses and investigate the possible sources of uncertainty in the simulated impacts.

## 2 Methods

### 2.1 RCM ensemble

Two experiments (GRASS and FOREST) were performed with an ensemble of nine RCMs whose names and characteristics are presented in Table 1. All experiments were performed at 0.44 degree (~50km) horizontal resolution on the EURO-CORDEX domain (Jacob et al. 2014) with lateral boundary conditions and sea surface temperatures prescribed based on 6-hourly ERA-Interim reanalysis (Dee et al. 2011). The simulations are analysed over the period 1986-2015 and the earlier years (1979-1985, or a subset of these years depending on models, see Table 1) were used as spin-up period. The model outputs

were aggregated to monthly values for use in this study.

A notable characteristic of the multi-model ensemble is that some RCMs share the same atmospheric scheme (i.e. same version and configuration) but are coupled to different Land Surface Models (LSMs), or share the same LSM in combination with different atmospheric schemes (see Table 1). This allows to evaluate the respective influence of atmospheric versus land processes representation. For instance, the same version of COSMO-CLM (CCLM) is used in combination with three different

LSMs (TERRA_ML, VEG3D and CLM4.5). Comparing results from these three CCLM-based configurations enables to isolate the role of land processes representation in this particular model. Conversely, CLM4.5 is used in combination with two different RCMs (CCLM and RegCM) which allows to diagnose the influence of atmospheric processes on the results. Different configurations of WRF are also used: WRFa-NoahMP and WRFb-NoahMP differ only in their atmospheric setup, while WRFb-NoahMP and WRFb-CLM3.5 share the same atmospheric setup but with different LSMs.




## 2.2 FOREST and GRASS vegetation maps

Two vegetation maps have been created for use in the Phase 1 LUCAS experiments (Fig. S1). The vegetation map used in experiment FOREST is meant to represent a theoretical maximum of tree coverage, while in the vegetation map used in experiment GRASS, trees are entirely replaced by grassland.

The starting point for both maps is a MODIS-based present-day land cover map at 0.5 degree resolution (Lawrence and Chase 2007) providing the global distribution of 17 Plant Functional Types (PFTs). Crops and shrubs which are present in the original map are not considered in the FOREST and GRASS experiments and are set to zero. To create the FOREST map, the fractional coverage of trees is expanded until trees occupy 100% of the non-bare soil area. The proportion of various tree types (i.e. broadleaf/needleleaf and deciduous/evergreen) is conserved as in the original map as well as the fractional coverage of bare

soil which prevents expanding vegetation on land areas where it could not realistically grow (e.g. in deserts). If no trees are present in a given grid cell with less than 100% bare soil, the zonal mean forest composition is taken as a representative value. This results in a map with only tree PFTs (PFT names) and bare soil, all other vegetation types being shrunk to zero. The GRASS map is then derived from the FOREST map by converting all tree PFTs into grassland PFTs, the C3 to C4 ratio being conserved as in the original MODIS-based map as well as the bare soil fraction.

Since the various RCMs use different land use classification schemes (see Table 1), the PFT-based FOREST and GRASS maps were converted into model-specific land use classes for implementation into the respective RCMs. The specific conversion rules used in each RCM are summarized in Table 1 (note that for three out of the nine RCMs no conversion was required). Urban areas, inland water and glacier, if included in a given RCM, were conserved as in the standard dataset of the respective RCM.


**Table 1: Names and characteristics of the RCMs used.**

| Model name | CCLM-TERRA | CCLM-VEG3D | CCLM-CLM4.5 | RCA | RegCM-CLM4.5 | REMO-iMOVE | WRFa-NoahMP | WRFb-NoahMP | WRFb-CLM3.5 |
|---|---|---|---|---|---|---|---|---|---|
| **Institute ID** | JLU/BTU/C MCC | KIT | ETH | SMHI | ICTP | GERICS | IDL | UHOH | AUTH |
| **RCM** | COSMO_5.0_clm9 | COSMO_5.0_clm9 | COSMO_5.0_clm9 | RCA4 | RegCM4.6.1 (Giorgi et al. 2012) | REMO2009 | WRF381 | WRF381 | WRF381 |
| **Land surface scheme** | TERRA-ML (Schrodin and Heise 2002) | VEG3D (Breil et al. 2018) | CLM4.5 (Oleson et al. 2013) | (Samuelsson et al. 2006) | CLM4.5 (Oleson et al. 2013) | iMOVE (Wilhelm et al. 2014) | NoahMP | NoahMP | CLM3.5 (Oleson et al. 2008) |
| **Land cover classes (classes effectively used in FOREST and GRASS in bold)** | **1: everg. br. forest** **2: desc. broad closed** **3: desc. br. open** **4: everg. Needleleaf forest** | **1: bare soil** 2: water 3: urban **4: deciduous forest** **5: coniferous forest** 6: mixed forest | **1: Bare Soil** **2: Needleleaf Evergreen Tree - Temperate** **3: Needleleaf Evergreen Tree - Boreal** | **1: bare soil** **2: open land** **3: needle leaf forest** **4: broad leaf forest** | **1: Bare Soil** **2: Needleleaf Evergreen Tree - Temperate** **3: Needleleaf Evergreen Tree - Boreal** | **1: tr. br. everg.** **2: tr. br. desc.** **3: temp. br. everg.** **4: temp. br. desc.** **5: everg. conif.** **6: desc.** | **1: everg. needleleaf** **2: desc. Needleleaf** 3: everg. Broadleaf **4: desc. Broadleaf** 5: mixed forests 6: closed | **1: everg. needleleaf** **2: desc. Needleleaf** 3: everg. Broadleaf **4: desc. Broadleaf** **5: mixed forests** 6: closed | **1: Bare Soil** **2: Needleleaf Evergreen Tree - Temperate** **3: Needleleaf Evergreen Tree - Boreal** |




| | | | | | | | | | |
|---|---|---|---|---|---|---|---|---|---|
| | **5: desc. needleleaf forest** 6: mixed leaf trees 7: fresh water flooded trees 8: saline water flooded trees 9: mosaic tree/natural veget. 10: burnt tree cover 11: everg. shupbs closed/open 12: desc. shrubs closed/open 13: herbac. veget. closed/open **14: grass** 15: flooded shrups or herbac. 16: cultivated and managed 17: mosaic crop/tree/net veget. 18: mosaic crop/shrub/grass **19: bare areas** 20: water 21: snow and ice 22. artificial surface 23: undefined | 7: cropland 8: special crops **9: grassland** 10: shrubland | **4: Needleleaf Deciduous Tree - Boreal** 5: Broadleaf Evergreen Tree - Tropical **6: Broadleaf Evergreen Tree - Temperate** 7: Broadleaf Deciduous Tree - Tropical **8: Broadleaf Deciduous Tree - Temperate** **9: Broadleaf Deciduous Tree - Boreal** 10: Broadleaf Deciduous Shrub - Temperate 11: Broadleaf Evergreen Shrub - Temperate 12: Broadleaf Deciduous Shrub - Boreal **13: C3 artic grass** **14: C3 grass** **15: C4 grass** 16: Crop 1 17: Crop 2 | | **4: Needleleaf Deciduous Tree - Boreal** 5: Broadleaf Evergreen Tree - Tropical **6: Broadleaf Evergreen Tree - Temperate** 7: Broadleaf Deciduous Tree - Tropical **8: Broadleaf Deciduous Tree - Temperate** **9: Broadleaf Deciduous Tree - Boreal** 10: Broadleaf Deciduous Shrub - Temperate 11: Broadleaf Evergreen Shrub - Temperate 12: Broadleaf Deciduous Shrub - Boreal **13: C3 artic grass** **14: C3 grass** **15: C4 grass** 16: Crop 1 17: Crop 2 | conif. 7: everg. shrubs 8: desc. shrubs **9: C3 grasses** **10: C4 grasses** 11: tundra 12: swamps 13: C3 crops 14: C4 crops 15: urban **16: bare** | shrubland 7: open shrubland 8: wooded savannah 9: savannah **10: grassland** 11: wetlands 12: cropland 13: urban and built-up 14: cropland/natural vegetation mosaic **15: snow and ice** **16: barren or sparsely vegetated** **17: water** 18: wooded tundra 19: mixed tundra 20: barren tundra 21: lakes | shrubland 7: open shrubland 8: wooded savannah 9: savannah **10: grassland** 11: wetlands 12: cropland 13: urban and built-up 14: cropland/natural vegetation mosaic **15: snow and ice** **16: barren or sparsely vegetated** **17: water** 18: wooded tundra 19: mixed tundra 20: barren tundra 21: lakes | **4: Needleleaf Deciduous Tree - Boreal** 5: Broadleaf Evergreen Tree - Tropical **6: Broadleaf Evergreen Tree - Temperate** 7: Broadleaf Deciduous Tree - Tropical **8: Broadleaf Deciduous Tree - Temperate** **9: Broadleaf Deciduous Tree - Boreal** 10: Broadleaf Deciduous Shrub - Temperate 11: Broadleaf Evergreen Shrub - Temperate 12: Broadleaf Deciduous Shrub - Boreal **13: C3 artic grass** **14: C3 grass** **15: C4 grass** 16: Crop1 17: Crop2 |
| **Conversion method to implement the PFT-based input vegetation maps (FOREST and GRASS)** | bare soil=19 Needleleaf evergreen tree-temperate=4 Needleleaf evergreen tree-boreal=4 Needleleaf deciduous tree- | bare soil=1 Needleleaf evergreen tree-temperate=5 Needleleaf evergreen tree-boreal=5 Needleleaf deciduous tree- | No conversion needed | Bare soil = 1 Needleleaf evergreen tree-temperate = 3 Needleleaf evergreen tree-boreal = 3 Needleleaf deciduous | No conversion needed | bare soil=16 Needleleaf evergreen tree-temperate=5 Needleleaf evergreen tree-boreal=5 Needleleaf deciduous tree- | Bare soil = 16 Needleleaf evergreen tree-temperate = 1 Needleleaf evergreen tree-boreal = 1 Needleleaf | Bare soil = 16 Needleleaf evergreen tree-temperate = 1 Needleleaf evergreen tree-boreal = 1 Needleleaf | No conversion needed |



| | | | | | | | | | |
|---|---|---|---|---|---|---|---|---|---|
| | boreal=5 Broadleaf evergreen tree - tropical=1 Broadleaf evergreen tree – temperate=1 Broadleaf deciduous tree – tropical=2 Broadleaf deciduous tree – temperate=2 Broadleaf deciduous tree – boreal=3 C3 arctic grass= 14 C3 grass= 14 C4 grass= 14 | boreal=5 Broadleaf evergreen tree - tropical=4 Broadleaf evergreen tree – temperate=4 Broadleaf deciduous tree – tropical=4 Broadleaf deciduous tree – temperate=4 Broadleaf deciduous tree – boreal=4 C3 arctic grass =9 C3 grass =9 C4 grass =9 | | tree-boreal = 3 Broadleaf evergreen tree -tropical = 4 Broadleaf evergreen tree – temperate = 4 Broadleaf deciduous tree –tropical = 4 Broadleaf deciduous tree – temperate = 4 Broadleaf deciduous tree –boreal = 4 C3 arctic grass = 2 C3 grass = 2 C4 grass = 2 | | boreal=6 Broadleaf evergreen tree - tropical=1 Broadleaf evergreen tree – temperate=3 Broadleaf deciduous tree – tropical=2 Broadleaf deciduous tree – temperate=4 Broadleaf deciduous tree – boreal=4 Broadleaf evergreen shrub – temperate=7 Broadleaf deciduous shrub – temperate=8 Broadleaf deciduous shrub – boreal=8 C3 arctic grass=9 C3 grass=9 C4 grass=10 | deciduous tree-boreal = 2 Broadleaf evergreen tree -tropical = 3 Broadleaf evergreen tree – temperate = 3 Broadleaf deciduous tree –tropical = 4 Broadleaf deciduous tree – temperate = 4 Broadleaf deciduous tree –boreal = 4 C3 arctic grass = 10 C3 grass = 10 C4 grass = 10 | deciduous tree-boreal = 2 Broadleaf evergreen tree -tropical = 3 Broadleaf evergreen tree – temperate = 3 Broadleaf deciduous tree –tropical = 4 Broadleaf deciduous tree – temperate = 4 Broadleaf deciduous tree –boreal = 4 C3 arctic grass = 10 C3 grass = 10 C4 grass = 10 | |
| **Representation of sub-grid scale vegetation heterogeneity** | Single class | Single class | Tile approach | Tile approach | Tile approach | Tile approach | Single class | Single class | Tile approach |
| **Leaf Area Index** | Prescribed seasonal cycle (sinus function depending on altitude and latitude with vegetation-dependent minimum and maximum values) | Prescribed seasonal cycle (sinus function depending on altitude and latitude with vegetation-dependent minimum and maximum values) | Prescribed seasonal cycle based on MODIS (Lawrence and Chase 2007) | Calculated monthly based on vegetation type, soil temperature and soil moisture | Prescribed seasonal cycle based on MODIS (Lawrence and Chase 2007) | Calculated daily based on atmospheric forcing and soil moisture state | Prescribed seasonal cycle based on lookup tables | Prescribed seasonal cycle based on lookup tables | Prescribed seasonal cycle based on MODIS (Lawrence and Chase 2007) |
| **Total soil depth and number of hydrologically /thermally active soil layers** | 9 thermally active layers down to 7.5 m; first 8 hydrological | 9 layers down to 7.5 m | 15 layers for thermal calculations down to 42 m; first 10 hydrologically active | 5 layers down to 2.89 m | 15 layers for thermal calculations down to 42 m; first 10 hydrologically active | 5 thermally active layers down to 10 m; 1 water bucket | 4 layers down to 1 m | 4 layers down to 1 m | 10 layers down to 3.43 m |



| | ly active down to 3.9 m | | down to 3.43 m | down to 3.43 m | | | | |
|---|---|---|---|---|---|---|---|---|
| **Initialisation and spin up** | Initialization with ERA-Interim, 1979-1985 as spin-up | Initialization with ERA-Interim, 1979-1985 as spin-up | Initialization with ERA-Interim, 1979-1985 as spin-up | Initialization with ERA-Interim, 1979-1985 as spin-up | Initialization with ERA-Interim except soil moisture which is based on a climatological average (Giorgi et al. 1989); 1985 as spin-up | Initialization with ERA-Interim, 1979-1985 as spin-up | Initialization with ERA-Interim, 1979-1985 as spin-up | Initialization with ERA-Interim, 1983-1985 as spin-up | Initialization with ERA-Interim, 1984-1985 as spin-up |
| **Lateral boundary formulation** | (Davies 1976) | (Davies 1976) | (Davies 1976) | (Davies 1976) with a cosine-based relaxation function | (Giorgi et al. 1993) | (Davies 1976) | exponential relaxation | exponential relaxation | expotential relaxation |
| **Buffer (No. of grid cells)** | 13 | 13 | 13 | 8 | 12 | 8 | 15 | 10 | 10 |
| **No. of vertical levels** | 40 | 40 | 40 | 24 | 23 | 27 | 50 | 40 | 40 |
| **Turbulence and planetary boundary layer scheme** | Level 2.5 closure for turbulent kinetic energy as prognostic variable (Mellor and Yamada 1982) | Level 2.5 closure for turbulent kinetic energy as prognostic variable (Mellor and Yamada 1982) | Level 2.5 closure for turbulent kinetic energy as prognostic variable (Mellor and Yamada 1982) | | The University of Washington turbulence closure model (Grenier et al. 2001; Bretherton et al. 2004) | Vertical diffusion after (Louis 1979) for the Prandtl layer, extended level-2 scheme after (Mellor and Yamada 1974) in the Ekman layer and the free atmosphere including modifications in the presence of clouds | MYNN Level 2.5 PBL (Nakanishi and Niino 2006; NAKANISHI and NIINO 2009) | MYNN Level 2.5 PBL (Nakanishi and Niino 2006; NAKANISHI and NIINO 2009) | MYNN Level 2.5 PBL (Nakanishi and Niino 2006; NAKANISHI and NIINO 2009) |
| **Radiation scheme** | (Ritter et al. 1992) | (Ritter et al. 1992) | (Ritter et al. 1992) | (Savijärvi and Savijärvi 1990), Wyser et al (1999) | Radiative transfer model from the NCAR Community Climate Model 3 (CCM 3) (Kiehl et al., 1996) | (Morcrette et al. 1986) with modifications for additional greenhouse gases, ozone and various aerosols. | Rapid Radiative Transfer Model (RRTMG) scheme (Iacono et al. 2008) | Rapid Radiative Transfer Model (RRTMG) scheme (Iacono et al. 2008) | Rapid Radiative Transfer Model (RRTMG) scheme (Iacono et al. 2008) |
| **Convection scheme** | (Tiedtke 1989) | (Tiedtke 1989) | (Tiedtke 1989) | (Bechtold et al. 2001) | (Tiedtke 1996) for cumulus convection | (Tiedtke 1989) with modifications after Nordeng (1994) | (Grell and Freitas 2014) for cumulus convection and Global/Regional | (Kain and Fritsch 1990); no shallow convection | (Kain and Fritsch 1990); no shallow convection |



| | | | | | | | | | |
|---|---|---|---|---|---|---|---|---|---|
| | | | | | | | Integrated Modeling System (GRIMS) Scheme (Hong et al. 2013) for shallow convection | | |
| **Microphysics scheme** | One-moment cloud microphysics scheme (Seifert and Beheng 2001) | One-moment cloud microphysics scheme (Seifert and Beheng 2001) | One-moment cloud microphysics scheme (Seifert and Beheng 2001) | Values from tables | Subgrid Explicit Moisture scheme (SUBEX) (Pal et al. 2000) | (Sundqvist 1978)(Roeckner et al., 1996) | Two-moment, 6-class scheme (Lim and Hong 2010) | (Thompson et al. 2004) | (Thompson et al. 2004) |
| **Greenhouse gases** | Historical | Historical | Historical | Historical | Historical | Historical | Historical | Constant | Constant |
| **Aerosols** | Constant (Tanré, 1984) | (Tegen et al. 1997) climatology | Constant (Tanré, 1984) | Constant | Not accounted for | Constant (Teichmann et al. 2013) | (Tegen et al. 1997) climatology | (Tegen et al. 1997) climatology | (Tegen et al. 1997) climatology |

## 3 Results

### 3.1 Temperature response

The effect of forestation (FOREST minus GRASS) on seasonal mean winter 2-meter temperature is shown in Figure 1. All RCMs simulate a warming pattern which is strongest in the northeast of Europe. This warming effect weakens toward the southwest of the domain to even become a cooling effect for instance in the Iberian Peninsula (except for REMO-iMOVE). In summer (Fig. 2), there is a very large spread of model responses with some RCMs predicting a widespread cooling from forestation (CCLM-TERRA and RCA), a widespread warming (RegCM-CLM4.5, REMO-iMOVE and the WRF models) or a mixed response (CCLM-VEG3D and CCLM-CLM4.5). This overall highlights the strong seasonal contrasts in the temperature effect of forestation and the larger uncertainties associated with the summer response.

Looking separately at the response for daytime and nighttime 2-meter temperatures also indicates important diurnal contrasts. The winter warming effect is stronger and more widespread for daily maximum temperature (Fig. 3), while daily minimum temperature shows a more contrasted cooling-warming dipole across the domain (Fig. 5). In summer, diurnal contrasts are even more pronounced with a majority of models showing an opposite sign of change for daily maximum and minimum temperatures over most of Europe (Fig. 4 and 6), namely a daytime warming effect and a nighttime cooling effect. Exceptions are RCA and CCLM-TERRA which indicate a cooling for both daily maximum and minimum temperatures and REMO-iMOVE exhibiting a warming for both daytime and nighttime.





In terms of magnitude, the temperature signal is substantial. In all RCMs, there is at least one season with absolute temperature changes above 2 degrees in some regions, for instance in winter and spring over Northern Europe. The magnitude of changes is even more pronounced for daily maximum temperature. Changes in surface (skin) temperature (Fig. S5) are overall broadly consistent with the above-mentioned results based on 2-meter temperature. Yet, differences in magnitude or even in sign of the signal occur locally in all models, indicating that the choice of temperature metric can have a critical role when assessing

local LUC impacts as noted in previous studies (Meier et al. 2018; Winckler et al. 2018).

## 3.2 Surface energy balance

Changes in surface energy fluxes over land are summarized for eight European regions (the Alps, the British Isles, Eastern Europe, France, the Iberian Peninsula, the Mediterranean, Mid-Europe and Scandinavia) as defined in the PRUDENCE project

(Christensen et al. 2007). Here we discuss results for two selected regions representative of Northern Europe (Scandinavia; Fig. 7) and Southern Europe (the Mediterranean; Fig. 8), while results for the full set of regions are provided in the Supplementary Information (S12 to S19). One of the most robust features across models and seasons is an increase in surface net shortwave radiation. This increase is a direct consequence of the impact of forestation on surface albedo. Indeed all RCMs consistently simulate a year-round decrease in surface albedo due to the lower albedo of forest compared to grassland (Fig S8).

This decrease is strongest in winter and at high latitudes owing to the snow masking effect of forest. However, the strongest increase in absorbed shortwave radiation occurs in spring and summer in both regions because incoming radiation is higher in these seasons, thus implying a larger surface radiation gain despite the smaller absolute change in albedo. Notable outliers are REMO-iMOVE, exhibiting a smaller albedo decrease across all seasons and thus a less pronounced increase in absorbed shortwave radiation, and CCLM-TERRA and RCA, which despite the albedo increase simulate a net shortwave radiation

decrease in summer (only over Scandinavia in the case of RCA). In the latter two models, an increase in evapotranspiration triggers an increase in cloud cover and a subsequent decrease in incoming shortwave radiation (not shown) offsetting the change in surface albedo.

To a large extent, sensible heat flux follows shortwave radiation changes (i.e. a majority of models suggest an increase in sensible heat). This is also largely the case for ground heat flux (calculated here indirectly as the residual of the surface energy

balance) which increases in autumn, winter and spring in most models due to the overall increase in absorbed radiation. Changes in the latent heat flux exhibit a higher degree of disagreement across models and seasons. For instance in spring, latent heat flux increases together with sensible heat over Scandinavia while it decreases in most models over the Mediterranean. In summer, the agreement is low over Scandinavia and there is a tendency for decreasing latent heat in the Mediterranean. We note that this response is opposite to the expectation that trees can sustain more evaporation during dry

conditions compared to grassland because of their deeper root system (Ellison et al. 2017; Meier et al. 2018).



### 3.3 Origin of the inter-model spread

Changes in albedo and in the partitioning of turbulent heat fluxes are essential in determining the temperature effect of
forestation. The warming influence of albedo decrease is for instance evident in winter and spring over Northern Europe as
seen in the previous section. The role of the turbulent heat fluxes partitioning can be illustrated by examining the link between
changes in 2-meter temperature and in evaporative fraction (EF), calculated as the ratio between latent heat and the sum of
latent and sensible heat. The advantage of using EF instead of latent heat flux is that the former provides a metric relatively
independent of albedo change (since albedo change does influence the magnitude of turbulent heat fluxes through changes in
available energy). Taking the example of Scandinavia in summer (Fig. 9), it appears that there is a relatively linear relationship
between changes in temperature and in EF. In other words, models showing a decrease in EF following forestation tend to
simulate a warming and models showing an increase in EF simulate a cooling. In order to assess more systematically the role
of albedo versus EF for different regions and seasons, we perform a multi-linear regression analysis using albedo and EF
changes as explaining variables. The variance explained (as indicated by the R-squared coefficients) by albedo and EF changes
is shown in Fig. 10 for Scandinavia and the Mediterranean. With the notable exception of the winter season, albedo and EF
combined are able to explain a large fraction of the inter-model variance of the simulated temperature response. Over
Scandinavia, albedo change explains the largest part of the inter-model variance in the spring temperature response, indicating
a dominance of radiative processes during this season. In summer and autumn, inter-model differences are mostly explained
by EF as is also the case in spring and autumn in the Mediterranean.

The rationale for using albedo and EF as main explaining factors is that they capture the intrinsic changes in land surface
characteristics brought about by LUC and represent respectively the associated radiative and non-radiative forcings. In fact,
albedo and EF combined effectively explain a large fraction of the inter-model variance of the simulated temperature response
as seen in Figure 10. However, it is important to note that this simplified approach ignores other factors that may contribute to
the temperature response (e.g. changes in surface roughness, atmospheric feedbacks). Most notably, in winter the inter-model
spread is not well explained by either albedo or EF, thus suggesting that other processes such as large-scale atmospheric
feedbacks may play a dominant role.

Comparing results from different RCMs sharing either the same LSM or the same atmospheric model can help provide
additional insights on the respective role of land versus atmospheric processes. Already by comparing the temperature signal
for different RCMs (Fig. 1 to 6), it appears, in summer particularly, that the three RCMs based on CCLM (i.e same atmospheric
model with three different LSMs) span almost the full range of RCM responses while CCLM-CLM4.5 and RegCM-CLM4.5
(i.e. same LSM and different atmospheric models) have generally similar patterns of change. This suggests that the summer
temperature response to forestation is conditioned primarily by land processes representation more than by atmospheric
processes. To quantify objectively the level of similarity or dissimilarity between different RCMs, we compute the Euclidean
distance across latitude and longitude between each RCM pairs for each season for differences in 2-meter temperature and
precipitation. This distance matrix is then used as a basis for a hierarchical clustering applying the Ward's clustering criterion





(Ward 1963). The cluster analysis indicates a relatively high degree of similarity in the temperature response for both winter and summer between CCLM-CLM4.5 and RegCM-CLM4.5 which share the same LSM (Fig. 11). On the other hand, the three CCLM-based RCMs are relatively far apart especially in summer despite the fact that they share a common atmospheric scheme. While this suggests the dominance of land processes in determining the temperature response to LUC, we note that
this is not confirmed by the three WRF-based configurations which tend to indicate a larger impact of atmospheric processes. Concerning the precipitation response, there is more clustering around atmospheric schemes in winter and a mixed picture in summer (Fig. 12). This indicates that, in particular in winter, atmospheric processes and feedbacks are more influential in determining precipitation changes, for which we note however no clear consensus across models (Fig. S6).

## 4 Discussion and Conclusions


Results from nine RCMs show that, compared to grassland, forests implies warmer temperatures in winter and spring over Northern Europe. This result is robust across RCMs and is a direct consequence of the lower albedo of forests which is the dominating factor during these seasons. In summer and autumn, however, the RCMs disagree on the direction of changes, with responses ranging from a widespread cooling to a widespread warming above 2 degrees in both cases. Although albedo change
plays an important role in all seasons by increasing absorbed surface radiation, in summer inter-model differences in the temperature response are to a large extent induced by differences in EF. These conclusions are overall consistent with previous studies based on global climate models. Results from the LUCID and the CMIP5 model intercomparisons have indeed highlighted a robust, albedo-induced, winter cooling effect due to past deforestation at mid-latitudes (Lejeune et al. 2017), in other words implying a winter warming effect of forestation. On the other hand, no robust summer response has been identified
in these intercomparisons, mainly attributed to a lack of agreement across models concerning evapotranspiration changes (De Noblet-Ducoudré et al. 2012; Lejeune et al. 2017, 2018).

Resolving this lack of consensus will require intensified efforts to confront models and observations and identify possible model deficiencies (Meier et al. 2018; Duveiller et al. 2018a; Boisier et al. 2013, 2014). For instance, a key feature emerging from observation-based studies is the fact that mid-latitude forests are colder during the day and warmer during the night
compared to grassland (Lee et al. 2011; Li et al. 2015; Duveiller et al. 2018b). It is striking that none of the LUCID and CMIP5 models reflect this diurnal behavior (Lejeune et al. 2017), nor do the RCMs analyzed in this study (i.e. a majority of RCMs have a diurnal signal opposite to observations, two other RCMs indicate a cooling effect of forests for both day and night, one exhibit a warming effect for both day and night). Similarly, the fact that a majority of RCMs simulate a decrease in evapotranspiration following forestation is at odds with current observational evidence (Chen et al. 2018; Meier et al. 2018;
Duveiller et al. 2018b) and might play a role in the simulated summer daytime warming in most RCMs. For instance, modifying evapotranspiration processes in CLM4.5 was found to improve the daytime temperature difference between grassland and



forest (Meier et al. 2018). An important insight from this first phase of RCM experiments is therefore that a particular attention should be given to model evaluation and benchmarking in future phases of the LUCAS initiative.

An additional insight from this study concerns the role of land versus atmospheric processes. Some of the participating RCMs
share the same atmospheric scheme (i.e. same version and configuration) but are coupled to different land surface models, or share the same land surface model in combination with different atmospheric schemes. This represents a unique opportunity to objectively determine the origin of uncertainties in the simulated response. For instance, we find that land processes representation is heavily involved in the large model spread in summer temperature response. The range of responses generated by using three different LSMs within the same atmospheric scheme (CCLM) is almost as large as the full model range in
summer. Supporting this conclusion, a simple regression-based analysis shows that, except in winter, changes in albedo and EF can explain most of the inter-model spread in temperature sensitivity, in other words indicating that land processes primarily determine the simulated temperature response. Atmospheric processes can nevertheless also play a substantial or even dominant role for example in winter or for other variables such as precipitation.

In this first phase of LUCAS, we relied on idealised experiments at relatively low resolution (50 km) to gain insights on the
biogeophysical role of forests across a range of European climates. Future phases of LUCAS will evolve toward increasing realism for instance by 1) investigating transient historical LUC forcing as well as RCP-based LUC scenarios, 2) considering a range of land use transitions beyond grassland to forest conversion and 3) assessing the added-value of higher (kilometre-scale) resolution when assessing local LUC impacts. Finally, the most societally-relevant adverse effects or benefits from land management strategies may become apparent only when addressing changes in extreme events such as heatwaves or droughts
(Davin et al. 2014; Lejeune et al. 2018), an aspect which will receive more attention in future analyses based on LUCAS simulations.

**Acknowledgements**

E.L. Davin acknowledges support from the Swiss National Science Foundation (SNSF) through the CLIMPULSE project and
thanks the Swiss National Supercomputing Centre (CSCS) for providing computing resources. R.M. Cardoso and P.M.M. Soares acknowledge the projects LEADING (PTDC/CTA-MET/28914/2017) and FCT- UID/GEO/50019/2019 - Instituto Dom Luiz. P. Hoffmann was partly supported by the HICSS (Helmholtz-Institut Climate Service Science) project LANDMATE. L. Jach, K. Warrach-Sagi and V. Wulfmeyer acknowledge support by the state of Baden-Württemberg through bwHPC and thank the Anton and Petra Ehrmann-Stiftung Research Training Group "Water-People-Agriculture" for financial
support. The work of E. Katragkou and G. Sofiadis was supported by computational time granted from the Greek Research & Technology Network (GRNET) in the National HPC facility - ARIS - under project ID pr005025_thin. N. de Noblet-Ducoudré thanks the "Investments d'Avenir" Programme overseen by the French National Research Agency (ANR) (LabEx BASC; ANR-11-LABX-0034). RCA simulations were performed on the Swedish climate computing resource Bi provided by the Swedish National Infrastructure for Computing (SNIC) at the Swedish National Supercomputing Centre (NSC) at Linköping
University. G. Strandberg was partly funded by a research project financed by the Swedish Research Council VR



(Vetenskapsrådet) on "Quantification of the biogeophysical and biogeochemical forcings from anthropogenic deforestation on regional Holocene climate in Europe, LandClim II". S. Strada has been supported by the TALENTS3 Fellowship Programme (FP code 1718349004) funded by the autonomous region Friuli Venezia Giulia via the European Social Fund (Operative Regional Programme 2014-2020) and administered by the AREA Science Park (Padriciano, Italy). CCLM-TERRA

simulations were performed at the German Climate Computing Center (DKRZ) through support from the Federal Ministry of Education and Research in Germany (BMBF). M.H. Tölle acknowledges the funding of the German Research Foundation (DFG) through grant 401857120. We thank Richard Wartenburger for providing the R scripts that have been used to perform the cluster analysis.

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





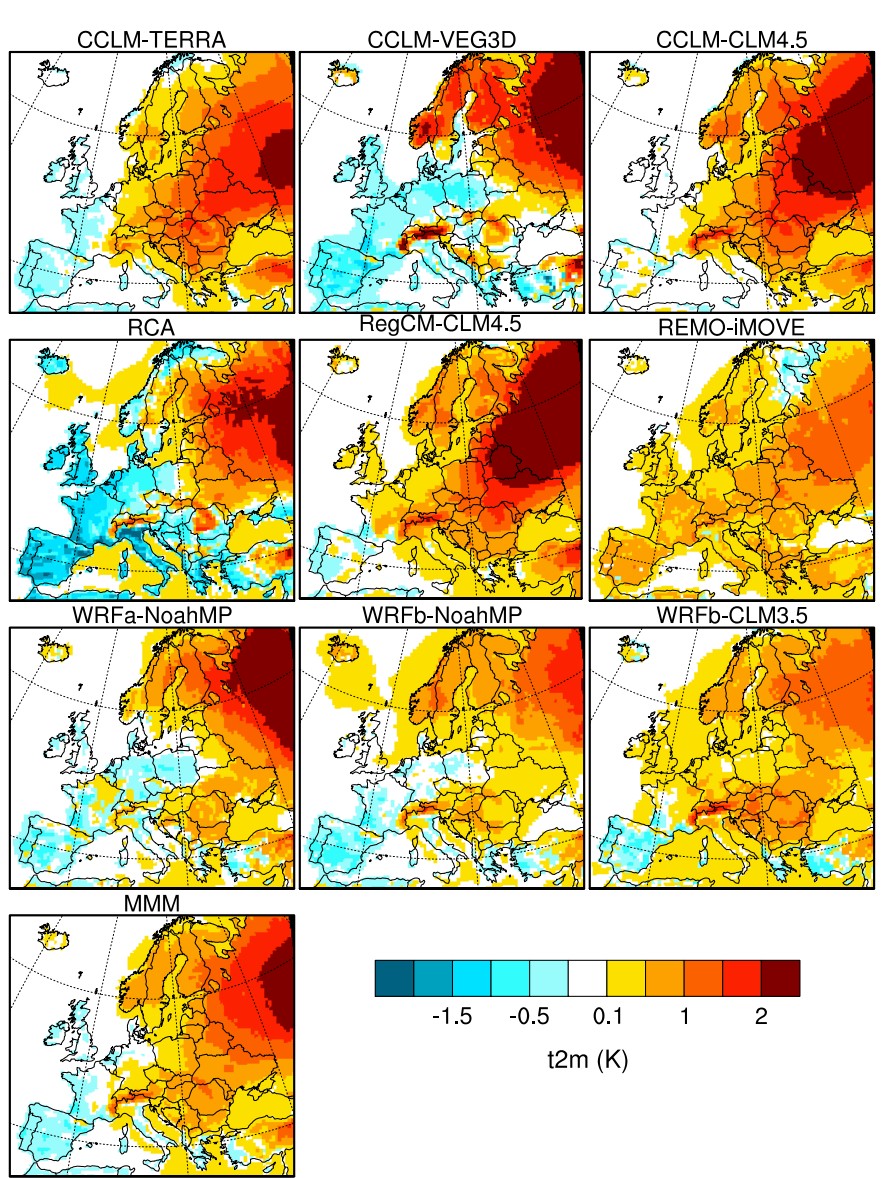

Figure 1: Seasonally-averaged 2-meter temperature difference (FOREST minus GRASS) for winter (DJF).





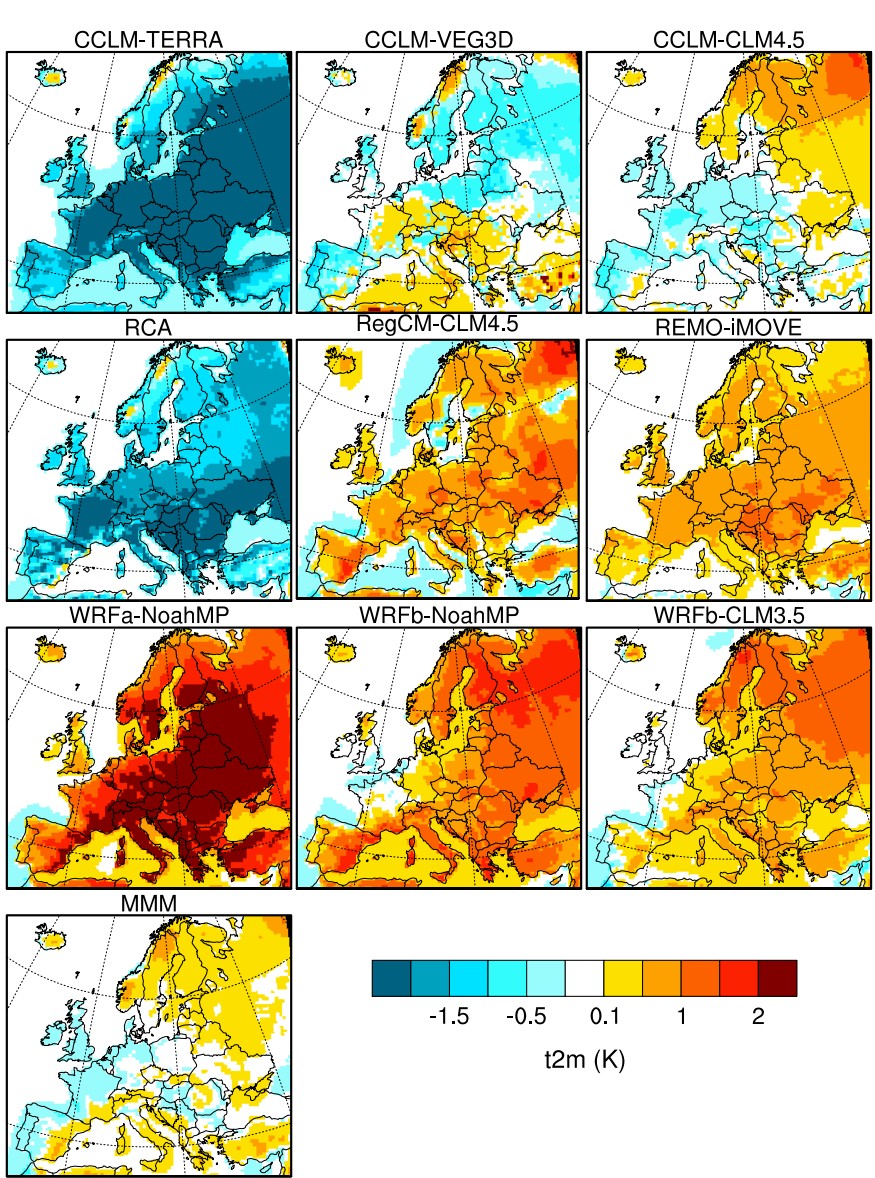

Figure 2: Seasonally-averaged 2-meter temperature difference (FOREST minus GRASS) for summer (JJA).





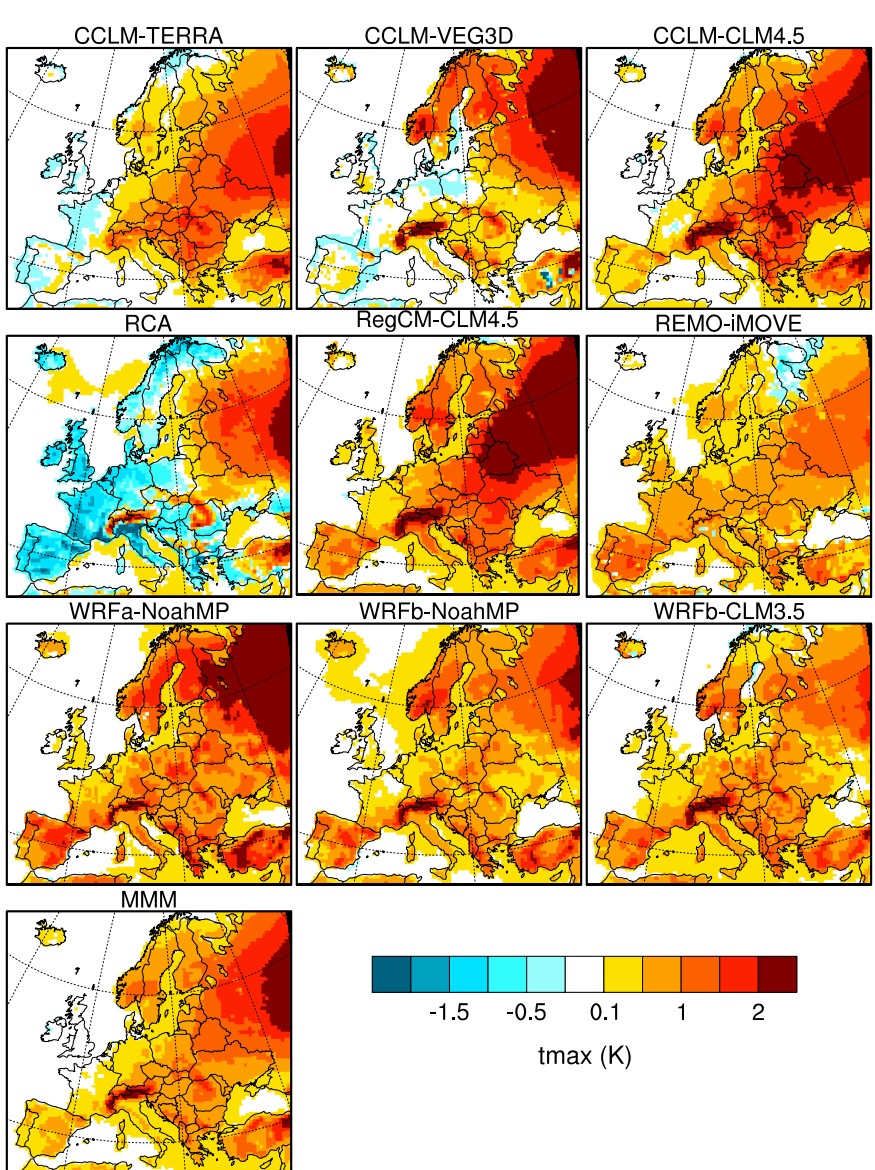

Figure 3: Seasonally-averaged daily maximum 2-meter temperature difference (FOREST minus GRASS) for winter (DJF).

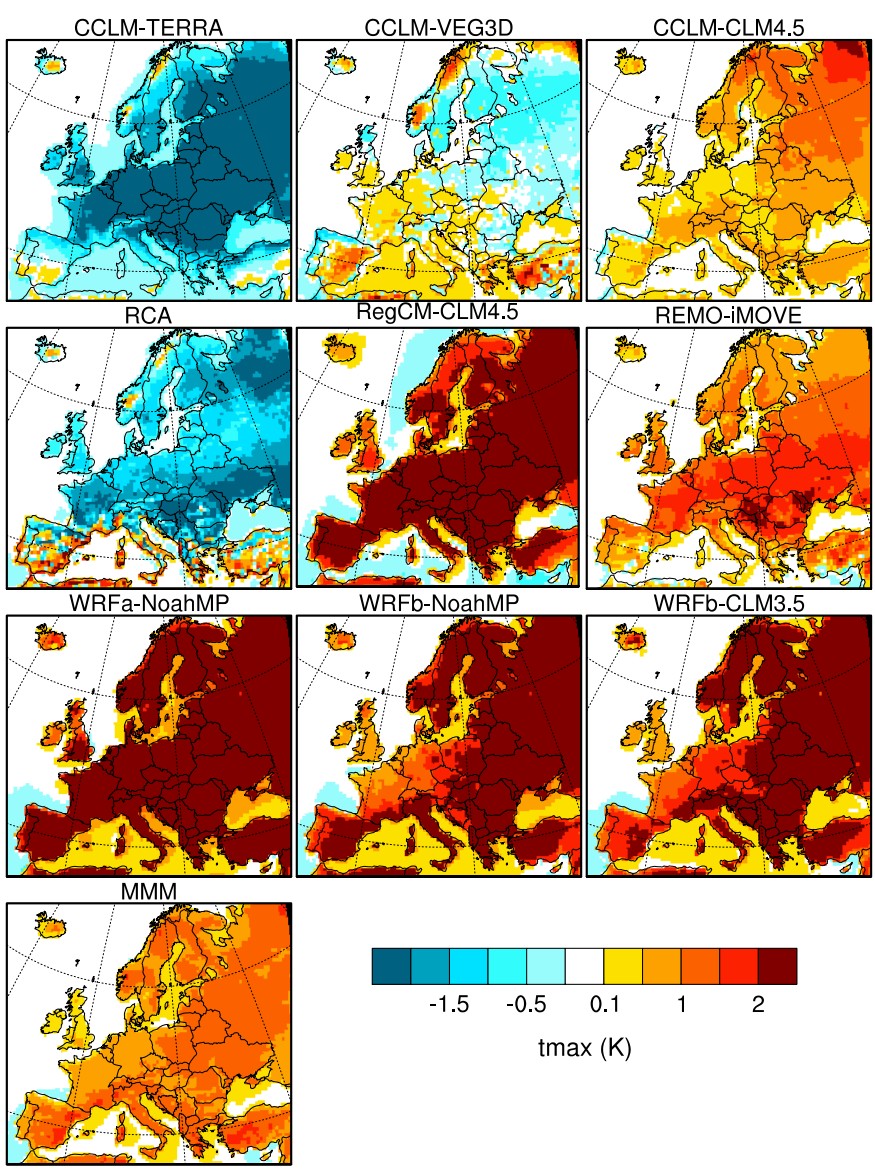

Figure 4: Seasonally-averaged daily maximum 2-meter temperature difference (FOREST minus GRASS) for summer (JJA).





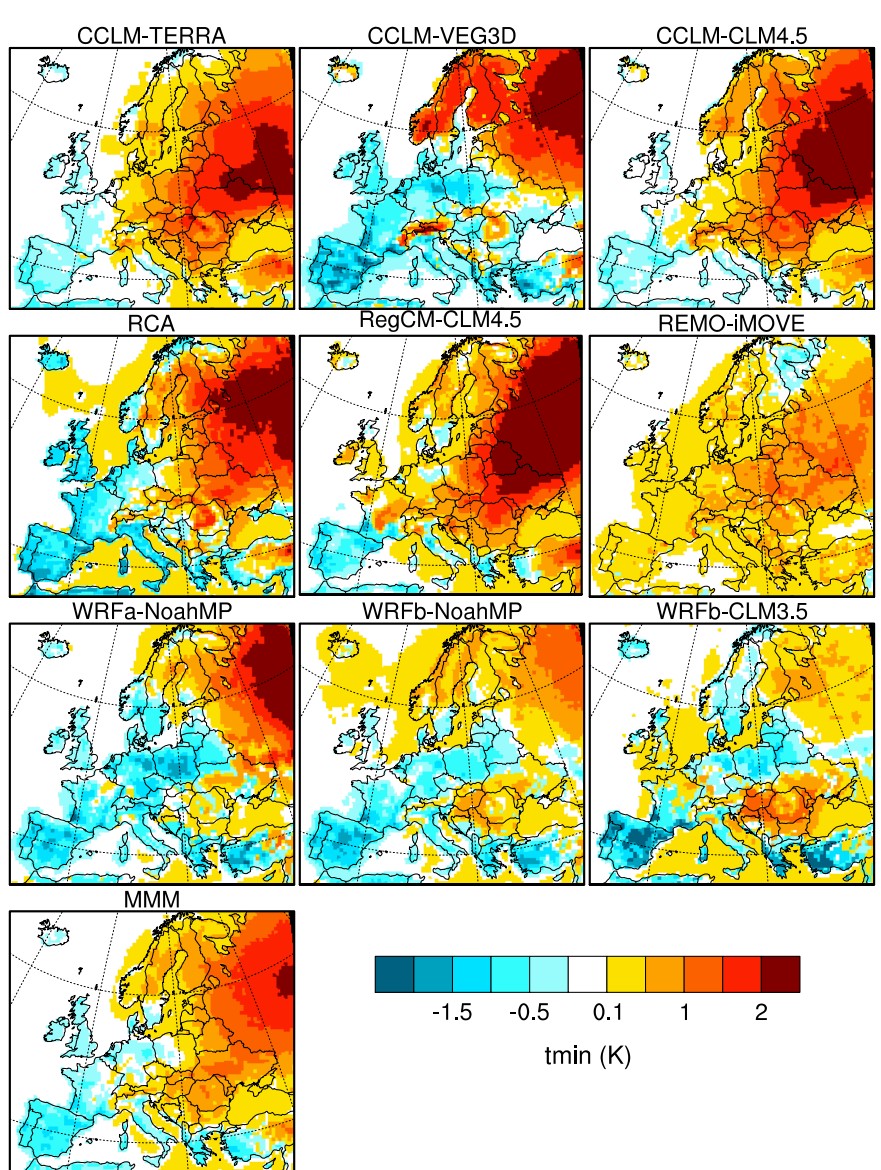

Figure 5: Seasonally-averaged daily minimum 2-meter temperature difference (FOREST minus GRASS) for winter (DJF).



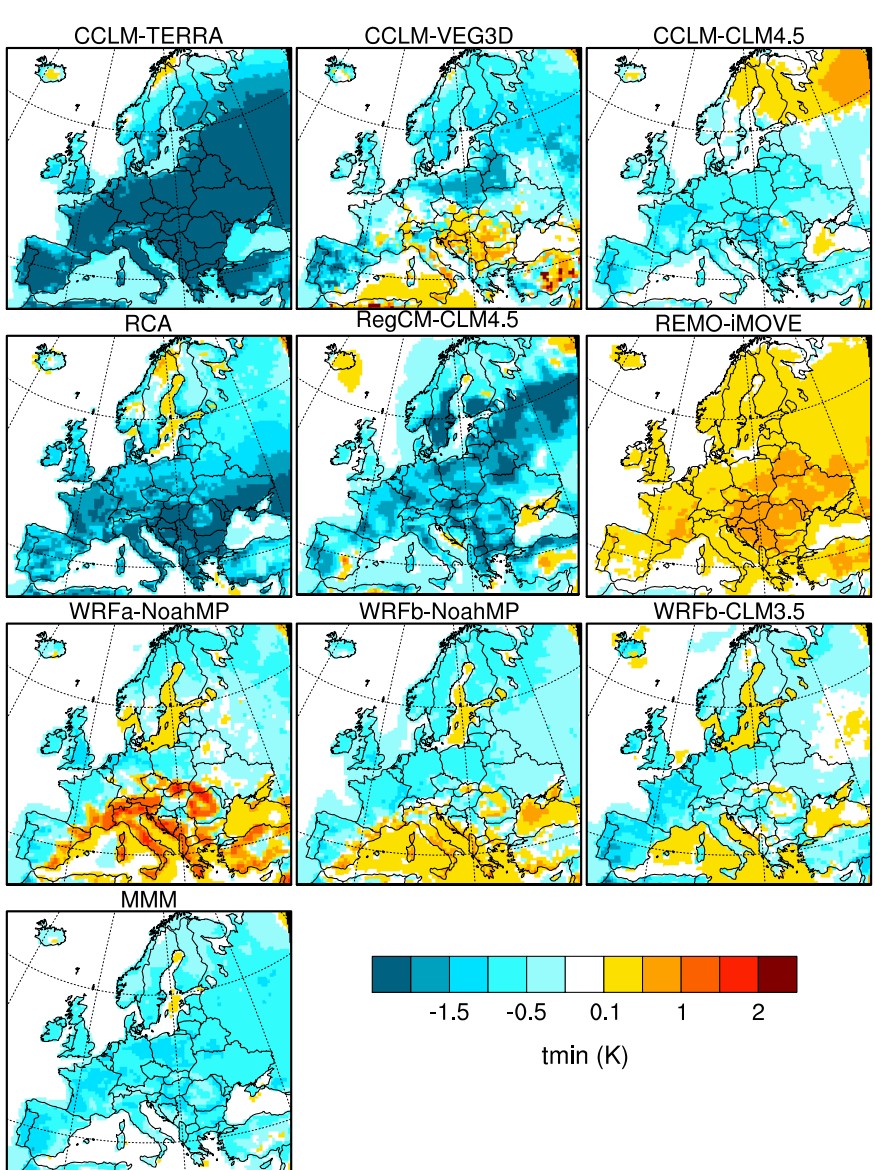

Figure 6: Seasonally-averaged daily minimum 2-meter temperature difference
(FOREST minus GRASS) for summer (JJA).





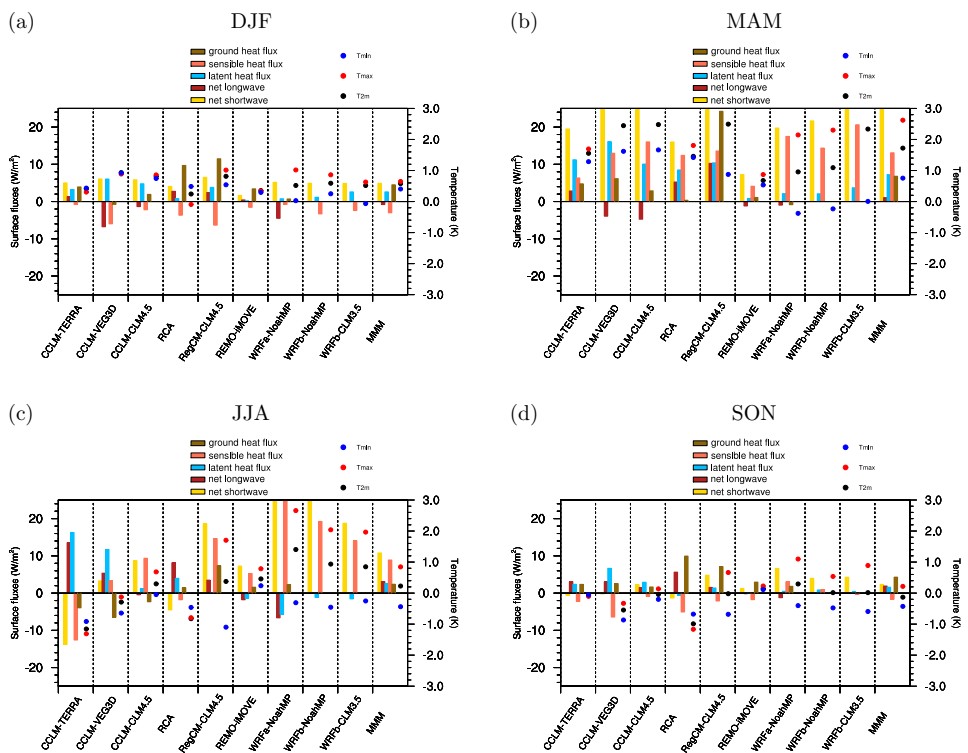

Figure 7: Changes in temperature and in surface energy balance components (FOREST minus GRASS) averaged over Scandinavia for DJF, MAM, JJA and SON. Results for other regions are shown in the appendix as well as difference maps for all variables.





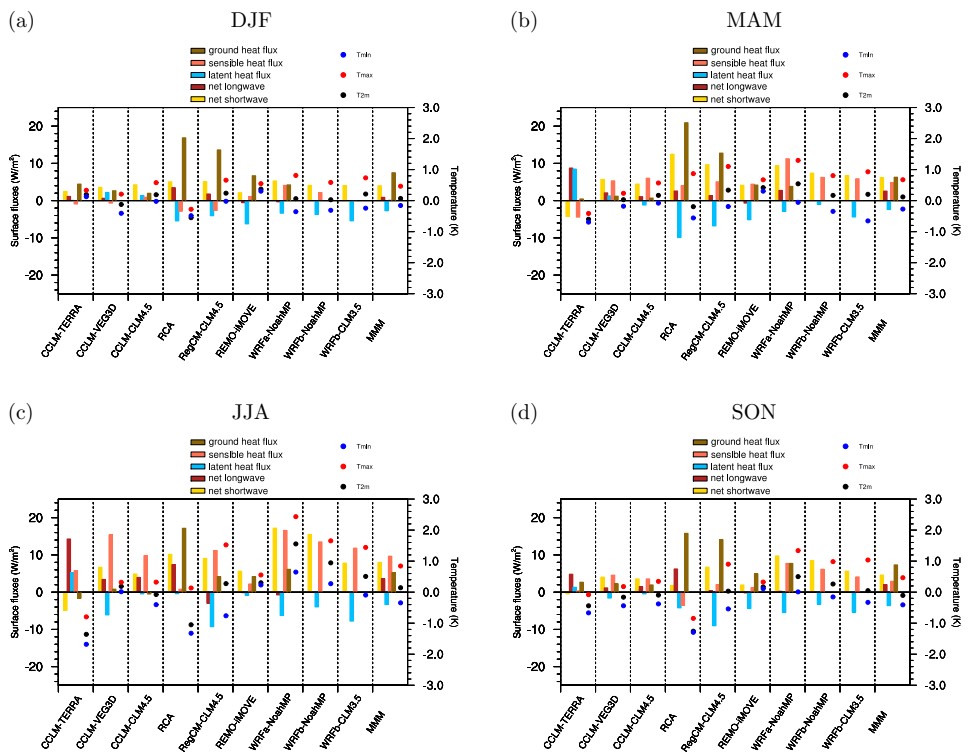

Figure 8: Changes in temperature and in surface energy balance components
(FOREST minus GRASS) averaged over the Mediterranean for DJF, MAM,
JJA and SON.



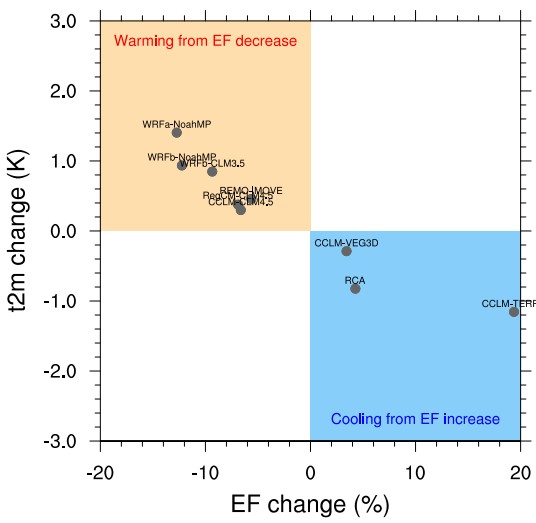

Figure 9: Relation between changes in 2-meter temperature and EF in summer over Scandinavia.





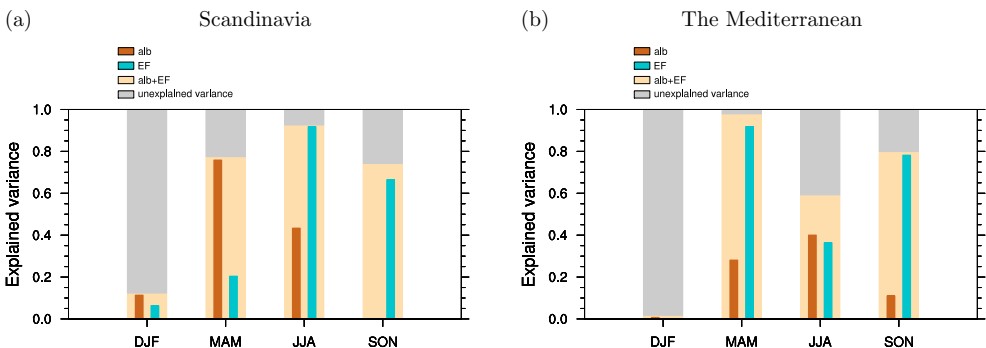

Figure 10: Fraction of inter-model variance in surface temperature difference explained by albedo change, evaporative fraction change or both combined. Alb: inter-model correlation (Rsquared) between albedo change and temperate change. EF: inter-model correlation (Rsquared) between albedo change and temperate change. Alb+EF: Rsquared of a multi-linear regression with both albedo change and EF change as predictors. Unexplained variance: variance explained by other factors than Alb+EF. Results for other regions are shown in the appendix.





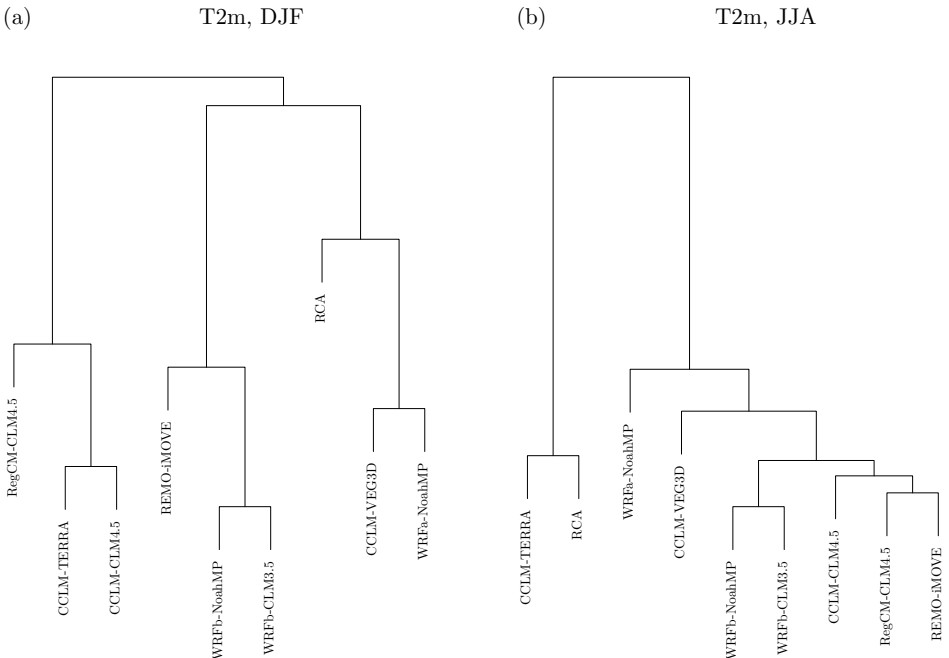

Figure 11: Dendrogram of the clustering analysis based on the 2-meter temperature response (FOREST minus GRASS) for DJF and JJA. The underlying distance matrix between RCM pairs is based on the Euclidean distance across latitude and longitude for the given season.





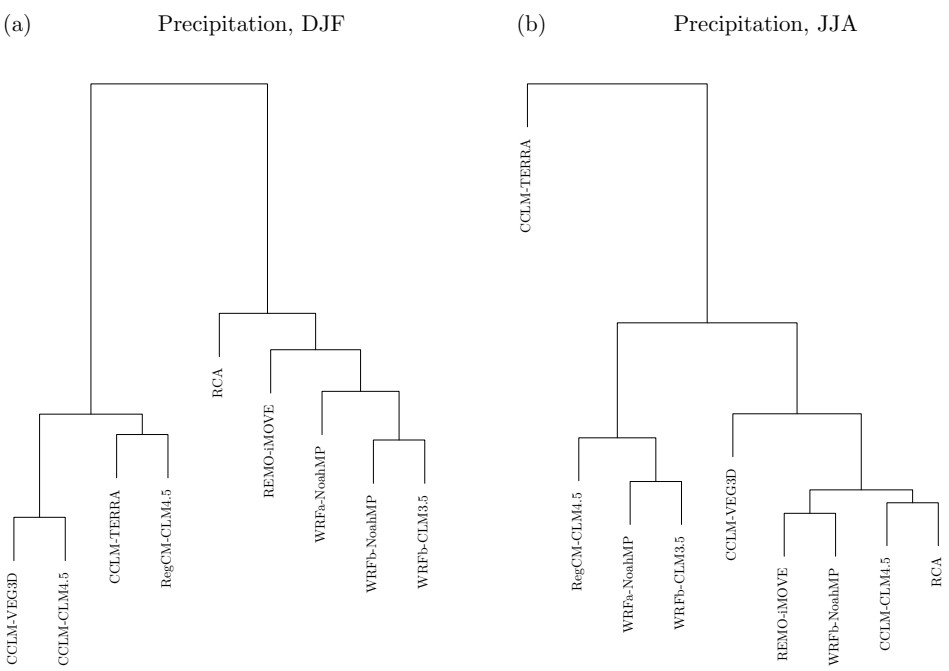

Figure 12: Same as previous figure but for precipitation.