# Peer review of "Biogeophysical impacts of forestation in Europe: First results from the LUCAS Regional Climate Model intercomparison"

_Earth System Dynamics, 2019_

## Referee Comment (RC1) · Anonymous Referee #1 · 8 Mar 2019

**General comments**

The study by Davin et al. seeks to understand the impact of vegetation cover across Europe, using multiple terrestrial biosphere models coupled with regional atmospheric models. To test the sensitivity of the coupled models, they carried out theoretical simulations using minimum (i.e. grassland) and maximum forest cover in the domain. The authors found that, except for a few consistent changes, such as increase in albedo in the grassland simulations, the response of the models diverged considerably, and seemed to be more related to differences amongst the terrestrial biosphere models than the atmospheric model. This study provides an interesting insight of the drivers

of divergent responses of coupled models to the same scenarios, and has a potential of becoming an important contribution to our understanding of biosphere-atmosphere interactions at regional scales. However, I think the manuscript needs some improvement and further development in the analysis and the interpretation of the results before it can be published.

First, the current analysis is a bit thin in assessing how atmospheric changes caused by the land cover scenarios are driving the near-surface variability across models. The authors briefly discuss these effects as potentially important, but this could be much more developed and quantified, considering that the authors are using coupled models. Just to cite one example, in the fraction of unexplained variance analysis (Fig. 10), the authors found that albedo and evaporative fraction show little explanatory power during the winter in Scandinavia. I think it should be straightforward to include some atmospheric variables such as cloud cover or incoming radiation, rainfall, or precipitable water as additional predictors. This analysis could help to quantify how surface and indirect atmospheric processes modulate surface temperature, and how this varies across regions and season.

In addition, as the authors noted in the text (Line 96), some combinations of models differ by only a few specific parametrisations. I think the authors could provide more insights on how the model settings and parameters are driving the differences in simulations. For example, between WRFa-NoahMP and WRFb-NoahMP simulations, the only relevant differences are the spin-up period and the sub-grid convection scheme, yet the results for precipitation in the summer are quite different (e.g Fig. 12). In this case, a brief discussion on how Grell and Freitas (2014) and Kain and Fritsch (1990) differ and how the changes in surface could impact the precipitation response given the assumptions of these schemes would be very informative. Likewise, the authors included CCLM simulations with three different land surface models and a similar discussion could be included.

Finally, the authors need to be more careful about the role of scale when discussing the

impacts of changes in land cover on the coupled biosphere-atmosphere system. For example, the authors stated that simulated changes in diurnal cycle due to forest/grass cover are the opposite of what previous observations suggest (near line 230). While I agree that this deserves further analysis, we cannot ignore that the impacts of replacing the vegetation of an entire continent on the diurnal cycle of temperature ought to be completely different from the impact of patchy deforestation/afforestation. At least in the tropics, the extent of deforestation in tropical forests may completely change the impact of land cover change in precipitation (e.g. Spracklen et al. 2018, doi:10.1146/annurev-environ-102017-030136, and references therein), and I would imagine that scale would also matter in higher latitudes and in other variables.

**Specific and minor comments**

Abstract. It would be helpful if the authors quantified their statements. For example, when they say that the albedo decreased with forestation, they could provide the range so readers would be able to judge whether the changes are important or not. This also applies to the range of temperature responses.

Discussion. I found somewhat hard to judge most of the results because little information is provided about how these different models usually perform across Europe. For instance, it would be very helpful to know whether some of them always underestimate rainfall or overestimate evapotranspiration, for example. Ideally the authors could show some model assessment with known benchmarks, but I understand that the model cannot be validated for idealised simulations. One suggestion would be to include one or two paragraphs describing previous studies using these models.

Line 36. Give concrete examples of atmospheric processes that dictate more directly the simulated response.

Line 51. A better justification is needed for the opening sentence. One could still see
some regional effects of land cover and land use change in global models, especially the extreme land cover scenarios used in this study.

Line 110-112. This is fine for the experiment, but the current approach does not completely prevent placing trees in areas naturally dominated by grasses or shrubs like some of the Mediterranean maquis.

Section 3.1. Figures 3-6 are interesting, but I think you could move some of them to the SI, and bring some of the atmospheric figures that could help explaining these differences to the main text (e.g. net radiation or precipitation).

Line 162. One interesting feature is that all CLM runs (CCLM, WRF, RegCM) seem to show increases in ET during the spring, but not during the summer. Could this be an extreme response to drought stress, like the beta factor being too low that stomata are closed for most of the summer?

Lines 165-170. Add references to figures/tables.

Lines 169-170. The last sentence belong to discussion instead of results, and I think this deserves to be expanded a bit more: this is likely to be related to my previous comment. One possibility could be that trees may be transpiring too much during the spring then running out of water during the summer. However, this is not so clear over land in Fig. S11. The strongest negative tendencies seem to be over the Mediterranean Sea and Black Sea. Just to confirm, do the averages in Figures 7–10 exclude grid cells over water?

Lines 185-189; lines 195-196; 212-213. If most of the variance in winter is not explained by albedo and evaporative fraction, then what is causing the variability? Could the variations be attributed to changes in weather patterns? The authors could and should include predictors that were not so surface-centric. The authors suggest that precipitation has no consensus amongst models (unsurprising, this is often more uncertain than other meteorological variables).

[Figure]

Lines 228-232: I do not necessarily see an inconsistency between model and independent observations. Europe is not a grassland continent nor a forest continent, and when an entire continent changes land cover, then the response will be very different from patches of forest next to patches of grassland that are side by side.

Line 242-243. I agree with this sentence, but the authors did not provide any insight on this. The authors could at least use the simulations in which differences are contained to indicate some of the origin of uncertainties. This would make the discussion much more interesting and informative, and go beyond the "models showed little or no agreement" storyline that we often see in model inter-comparison papers.

Table 1.

- I found this table a bit hard to read, I wonder if it would make it easier to separate the atmospheric and land surface settings.

- Was there any reason why the spin-up period was different for the three WRF simulations?

- In the greenhouse gas row, What is the difference between historical and constant? I assume historical means time varying, but this should be clarified, and a citation should be provided. For the simulations with constant values, provide these values, at least for $CO_2$.

Figures 1-6. What is MMM? I assume that it is the average response. Explain this in the figure captions.

Figure 9. Consider adding a similar panel for the Mediterranean.

---

## Referee Comment (RC2) · Anonymous Referee #2 · 22 Apr 2019

The paper addresses relevant scientific questions. There was a strong effort of several groups in order to provide modeling results and evaluate the modeling physics of several systems. The inter-comparison of the surface and atmospheric modeling systems and the two experiments (i.e. total FOREST x GRASS) provides an interesting tool for modeling improvement, mainly on the understanding the LSM impacts and feedbacks. The modeling results provide a new original contribution. The methodology is appropriated for the goals of the project. It provides variable Land Surface Systems and various atmospheric options. The paper structure has a good flow and fluency. See some questions & suggestions below. The results show that there were several differences among the modeling systems used on the inter-comparison. I think on the

methodology more information should be provided by the authors, mainly on the soil types and vegetation parameters such as the maximum/minimum stomatal resistance, vegetation height, and roots depth. Those parameters could allow improvements on the major conclusions. Furthermore, it could help the reproduction of the numerical experiments. The results discussions on the maps of temperature, don't provide analysis of map MMM (e.g. Figs 1 to 6). What are those maps? Mean of all modeling systems? Precipitation maps should be better analyzed in order to improve the conclusions on the temperature fields. Surface changes are likely to change the precipitation fields and as a feedback, the precipitation distribution is likely to have an impact the balance of radiation (downward and upward) and surface conditions, such as soil moisture and temperature. Furthermore, maps of precipitation could help on the interpretation of the discrepancies among models because this variable controls the vegetation transpiration, downward and upward short wave radiation, among others. Forest versus grass simulations for the Amazon, for instance, shows a strong change on the precipitation distribution caused by the transpiration (e.g. Ramos-da-Silva et al., J. Climate 2008). The authors should provide some insights on: how the forestation affects the major synoptic systems that move across Europe? Are these atmospheric systems enhanced or weakened? To improve the results analysis and discussion, known LSM model bias from previous studies could help on the results interpretation (e.g. Chen et al., JGR 2014). Some figures should be improved. Better legends could help the readers to quickly understand the presented images. For instance, what is MMM on the maps? Furthermore, in some figures, the fonts needs to be higher to permit a better reading (e.g. Figures 7, 8, 9 and 10). Figure 7 should have a higher threshold for net radiation. It is not clear the maximum on some cases. Further minor text corrections: Table 01 – Lateral boundary in the last column should be exponential (not expotential) Discussion – line 235-236 should be evapotranspiration (not evaporation).

---

## Author Comment (AC1) · 30 Jun 2019

We wish to thank the anonymous reviewers for their comprehensive and constructive comments. Below, we provide detailed responses and describe the corresponding changes in the manuscript.

*Referee #1*

*The study by Davin et al. seeks to understand the impact of vegetation cover across Europe, using multiple terrestrial biosphere models coupled with regional atmospheric models. To test the sensitivity of the coupled models, they carried out theoretical simulations using minimum (i.e. grassland) and maximum forest cover in the domain. The authors found that, except for a few consistent changes, such as increase in albedo in the grassland simulations, the response of the models diverged considerably, and seemed to be more related to differences amongst the terrestrial biosphere models than the atmospheric model. This study provides an interesting insight of the drivers of divergent responses of coupled models to the same scenarios, and has a potential of becoming an important contribution to our understanding of biosphere-atmosphere interactions at regional scales. However, I think the manuscript needs some improvement and further development in the analysis and the interpretation of the results before it can be published.*

*First, the current analysis is a bit thin in assessing how atmospheric changes caused by the land cover scenarios are driving the near-surface variability across models. The authors briefly discuss these effects as potentially important, but this could be much more developed and quantified, considering that the authors are using coupled models. Just to cite one example, in the fraction of unexplained variance analysis (Fig. 10), the authors found that albedo and evaporative fraction show little explanatory power during the winter in Scandinavia. I think it should be straightforward to include some atmospheric variables such as cloud cover or incoming radiation, rainfall, or precipitable water as additional predictors. This analysis could help to quantify how surface and indirect atmospheric processes modulate surface temperature, and how this varies across regions and season.*

Response: Thank you for this suggestion. It is correct that our findings suggest that atmospheric feedbacks are more important during winter and we agree that extending our correlation analysis to atmospheric variables could help demonstrating this point more directly. Note also that this study is meant to present a first overview of the LUCAS phase 1 results focusing on temperature and energy balance at seasonal timescales (as stated in the introduction) and will be complemented by further studies analyzing more specific aspects (in particular one study investigating the role of atmospheric processes in more details is planned).

Changes to manuscript: We will add incoming radiation in our correlation analysis. More generally we will add more description of changes in atmospheric variables and move some of the atmospheric figures from the SI into the main text (see below) in order to explicitly show the role of atmospheric processes.

*In addition, as the authors noted in the text (Line 96), some combinations of models differ by only a few specific parametrisations. I think the authors could provide more insights on how the model settings and parameters are driving the differences in simulations. For example, between WRFa-NoahMP and WRFb-NoahMP simulations, the only relevant differences are the spin-up period and the sub-grid convection scheme, yet the results for precipitation in the summer are quite different (e.g Fig. 12). In this case, a brief discussion on how Grell and Freitas (2014) and Kain and Fritsch (1990) differ and how the changes in surface could impact the precipitation response given the assumptions of these*

*schemes would be very informative. Likewise, the authors included CCLM simulations with three different land surface models and a similar discussion could be included.*

Response: The main advantage of presenting results from both WRFa-NoahMP and WRFb-NoahMP is that we can clearly attribute differences in simulated response to atmospheric processes (because the same LSM is used in both configurations). However, it is really challenging to attribute differences to specific atmospheric processes because the atmospheric schemes differ in several aspects (most notably convection and microphysics). As an example, the Kain-Fritsch scheme uses a mass flux approach which rearranges each column's mass in order to remove at least 90% of the convective available potential energy (CAPE). KF is highly influenced by the boundary layer forcing, particularly surface convergence (Gallus, 1999 and Liang et al., 2004b). The Grell-Freitas scheme is based on a scale-aware method with a large-scale instability tendency closure and is more sensitive to large-scale vertical motion than the KF scheme (Dai et al., 1999). Overall, the GF scheme is known to be usually drier than the KF scheme (Hu et al., 2018). In this light, it is well possible that the slight summer precipitation decrease in WRFa-NoahMP is related to the use of the GF scheme (while precipitation is less clearly affected in WRFb-NoahMP and WRFb-CLM3.5 which use the KF scheme). It is however not possible to do more than just speculating without dedicated sensitivity experiments with WRF which would be beyond the scope of this study. Concerning the simulations based on CCLM, a recent study has examined the sensitivity to land cover changes in CLM4.5 (Meier et al., 2018) and can indeed provide insights on the interpretation of the differences seen between CCLM-CLM4.5 and the other CCLM configurations (see also response below).

Changes to manuscript: We will add more discussion on the differences between model configurations and how this might relate to differences in the simulated response.

*Finally, the authors need to be more careful about the role of scale when discussing the impacts of changes in land cover on the coupled biosphere-atmosphere system. For example, the authors stated that simulated changes in diurnal cycle due to forest/grass cover are the opposite of what previous observations suggest (near line 230). While I agree that this deserves further analysis, we cannot ignore that the impacts of replacing the vegetation of an entire continent on the diurnal cycle of temperature ought to be completely different from the impact of patchy deforestation/afforestation. At least in the tropics, the extent of deforestation in tropical forests may completely change the impact of land cover change in precipitation (e.g. Spracklen et al. 2018, doi:10.1146/annurev-environ-102017-030136, and references therein), and I would imagine that scale would also matter in higher latitudes and in other variables.*

Response: We totally agree that the issue of scale might be an important aspect here. Observational methods capture mainly local changes in surface energy balance and temperature due to land cover and are unlikely to reflect the type of larger scale atmospheric feedbacks that can be triggered in coupled models (especially given the drastic land cover change imposed). In other words, the apparent discrepancy between models and observations may in part arise from the fact that models and observations differ in the scale of processes considered (and we should therefore be more careful not to attribute this discrepancy only to a lack of model realism). We agree that it is important to emphasize this point more clearly in the paper.

Changes to manuscript: While we will still keep the discussion on observation-based evidence because it provides an important context to our study, we will make sure to mention more explicitly the challenges arising from the issue of scale when comparing models and observations. We will also

emphasize the need for dedicated benchmarking efforts to tackle the issue of models to observations comparison.

*Specific and minor comments*
*Abstract. It would be helpful if the authors quantified their statements. For example, when they say that the albedo decreased with forestation, they could provide the range so readers would be able to judge whether the changes are important or not. This also applies to the range of temperature responses.*

Changes to manuscript: agreed, ranges of values will be added to the abstract

*Discussion. I found somewhat hard to judge most of the results because little information is provided about how these different models usually perform across Europe. For instance, it would be very helpful to know whether some of them always underestimate rainfall or overestimate evapotranspiration, for example. Ideally the authors could show some model assessment with known benchmarks, but I understand that the model cannot be validated for idealised simulations. One suggestion would be to include one or two paragraphs describing previous studies using these models.*

Response: As noted by the reviewer, the type of extreme land cover experiments performed here are not suitable for classical evaluation purpose. However, most of the RCMs used in our study have been part of EURO-CORDEX and have been evaluated in this context (e.g. Kotlarski et al 2014; Davin et al. 2016). Although for a given RCM the version and configuration used in our study may differ from the published EURO-CORDEX counterpart, the systematic biases that have been previously identified are still relevant here (e.g. predominant cold and wet biases for most European regions with the exception of Southern Europe in summer where the opposite occurs). We agree that this context should be provided.

Changes to manuscript: We will add a paragraph describing evaluation results from previous studies and the possible implications for our results.

*Line 36. Give concrete examples of atmospheric processes that dictate more directly the simulated response.*

Changes to manuscript: agreed, this will be added and the correlation analysis will be extended (see also response above).

*Line 51. A better justification is needed for the opening sentence. One could still see some regional effects of land cover and land use change in global models, especially the extreme land cover scenarios used in this study.*

Response: It is true that global models can also represent some regional LUC effects as we indeed summarize in the first paragraph. We are certainly not arguing that LUC should be included in RCMs rather than in GCMs, obviously it should be considered in both types of models. But the current situation is that RCMs intercomparison projects typically ignore LUC effects. We are therefore arguing that this should be remedied, one of the added values being (beside improving the consistency in forcings included in global and regional experiments) that new insights could be gained given the higher resolution at which RCMs operate (e.g. enabling to capture such local to regional effects in more details). While one could argue that this resolution issue is not critical in the extreme scenarios analyzed in the phase 1 of LUCAS, this nevertheless provides an essential motivation for the LUCAS project in general and beyond phase 1.

Changes to manuscript: we will clarify this argument

*Line 110-112. This is fine for the experiment, but the current approach does not completely prevent placing trees in areas naturally dominated by grasses or shrubs like some of the Mediterranean maquis.*

Response: It is certainly true that the FOREST map does not represent a potential vegetation map. Our intention was indeed to generate a map of maximum forest coverage, which we thought was more meaningful to explore the full forestation potential. There are already many examples of places in Europe where trees are growing where they would not be the naturally occurring vegetation type simply because of human intervention (assisted afforestation, forest management, fire suppression, etc). Our forest map is therefore less conservative in terms of potential for tree expansion than a potential vegetation map, which is in line with the idea of considering both reforestation and afforestation potential (note that we still exclude forest expansion over drylands that would likely imply drastic irrigation measures).

Changes to manuscript: we will clarify this point since our intention was indeed not explained well enough.

*Section 3.1. Figures 3-6 are interesting, but I think you could move some of them to the SI, and bring some of the atmospheric figures that could help explaining these differences to the main text (e.g. net radiation or precipitation).*

Changes to manuscript: We will move some of the atmospheric figures into the main text (but keep 3-6 as they are useful and described in the text).

*Line 162. One interesting feature is that all CLM runs (CCLM, WRF, RegCM) seem to show increases in ET during the spring, but not during the summer. Could this be an extreme response to drought stress, like the beta factor being too low that stomata are closed for most of the summer?*

Response: This behavior has been analyzed in details in an offline study with CLM4.5 (Meier et al., 2018) which indeed concluded that there is a too strong water limitation effect (too low beta factor in forest compared to open land) in summer in CLM4.5, while ET is higher in spring under forested conditions. The fact that this occurs also in the context of offline simulations confirm that this is an intrinsic feature of the CLM land surface model. Meier et al., 2018 tested various modifications to alleviate this issue but these modifications were not included in the coupled RCMs which are based on the default version of CLM.

Changes to manuscript: we will include a discussion on this particular behavior in CLM4.5

*Lines 165-170. Add references to figures/tables.*

Changes to manuscript: ok, references to figures will be added

*Lines 169-170. The last sentence belong to discussion instead of results, and I think this deserves to be expanded a bit more: this is likely to be related to my previous comment. One possibility could be that trees may be transpiring too much during the spring then running out of water during the summer. However, this is not so clear over land in Fig. S11. The strongest negative tendencies seem to be over*

*the Mediterranean Sea and Black Sea. Just to confirm, do the averages in Figures 7–10 exclude grid cells over water?*

Changes to manuscript: This sentence will be moved to the discussion and expended. In particular the discussion will be complemented by discussing the results of Meier et al., 2018 (see above). We will clarify that the average values indeed include only land points.

*Lines 185-189; lines 195-196; 212-213. If most of the variance in winter is not explained by albedo and evaporative fraction, then what is causing the variability? Could the variations be attributed to changes in weather patterns? The authors could and should include predictors that were not so surface-centric. The authors suggest that precipitation has no consensus amongst models (unsurprising, this is often more uncertain than other meteorological variables).*

Response: It would be beyond the scope of this manuscript to analyze weather patterns given the focus on the mean seasonal response in temperature and surface energy balance (weather phenomena would require outputs at much higher temporal resolution). But we will expand the discussion on the role of atmospheric processes as already discussed above.

Changes to manuscript: add atmospheric variables in the analysis of variance (see above)

*Lines 228-232: I do not necessarily see an inconsistency between model and independent observations. Europe is not a grassland continent nor a forest continent, and when an entire continent changes land cover, then the response will be very different from patches of forest next to patches of grassland that are side by side.*

Response: We agree that the scale effect can play an important role here (see response above)

Changes to manuscript: we will reformulate this part of the discussion as proposed above.

*Line 242-243. I agree with this sentence, but the authors did not provide any insight on this. The authors could at least use the simulations in which differences are contained to indicate some of the origin of uncertainties. This would make the discussion much more interesting and informative, and go beyond the "models showed little or no agreement" storyline that we often see in model intercomparison papers.*

Response: This sentence is about differences between surface versus 2-meter temperature. We agree that we did not give a lot of insights on this point (other than showing maps of surface temperature changes in the SI), but a separate study is currently being developed to explore the issue of surface versus 2-meter temperature in details (Breil et al., in prep).

Changes to manuscript: We will remove results based on surface temperature in the paper (i.e. SI fugure) and we will use only T2m for clarity and consistency across the manuscript.

*Table 1.*
*• I found this table a bit hard to read, I wonder if it would make it easier to separate the atmospheric and land surface settings.*
*• Was there any reason why the spin-up period was different for the three WRF simulations?*

*• In the greenhouse gas row, What is the difference between historical and constant? I assume historical means time varying, but this should be clarified, and a citation should be provided. For the simulations with constant values, provide these values, at least for CO2.*

Response: We find it convenient to keep all the settings in only one table, but we do list land settings first then atmospheric settings. We will try to find ways to make the table more readable. The spin-up in WRFb-Noah and WRFb-CLM3.5 is shorter due only to computational constraints. We will provide the constant values and clarify that historical means transient.

Changes to manuscript: The table will be clarified as suggested

*Figures 1-6. What is MMM? I assume that it is the average response. Explain this in the figure captions.*

Changes to manuscript: We will add the missing information: Multi-Model Mean (MMM)

*Figure 9. Consider adding a similar panel for the Mediterranean.*

Response: Thank you for this suggestion. We will consider adding a second panel to illustrate either another illustrative region or another explaining variable.

*Referee #2*

*The paper addresses relevant scientific questions. There was a strong effort of several groups in order to provide modeling results and evaluate the modeling physics of several systems. The intercomparison of the surface and atmospheric modeling systems and the two experiments (i.e. total FOREST x GRASS) provides an interesting tool for modeling improvement, mainly on the understanding the LSM impacts and feed-backs. The modeling results provide a new original contribution. The methodology is appropriated for the goals of the project. It provides variable Land Surface Systems and various atmospheric options. The paper structure has a good flow and fluency. See some questions & suggestions below.*

*The results show that there were several differences among the modeling systems used on the inter-comparison. I think on the methodology more information should be provided by the authors, mainly on the soil types and vegetation parameters such as the maximum/minimum stomatal resistance, vegetation height, and roots depth. Those parameters could allow improvements on the major conclusions. Furthermore, it could help the reproduction of the numerical experiments.*

Response: It is certainly conceivable that vegetation parameters could explain some of the differences between models. However, we don't think that listing these parameters will significantly shed light on the potential sources of model differences (mainly because it will not be possible to disentangle the effect of these parameters from the effect of other parameters/parameterization). That said, one particular feature of our RCM ensemble is that some RCMs are used in different configurations with a limited set of differences (e.g. WRF family) which can be an advantage when trying to attribute differences in model response to specific processes. We will therefore strengthen the discussion around specific parameterization differences in these models in order to make a clearer link between process representation and differences in simulated response (see also response to reviewer 1). Concerning specifically the role of vegetation parameters, a recent study using CLM4.5, which is also use here in two RCMs, investigated the role of vegetation/root parameters and water uptake parameterization on

the simulated effect grass to forest conversion (Meier et al., 2018). Since these results can help understand the behavior of the RCMs using CLM we will provide a discussion with a link to this study.

Changes to manuscript: We will add more discussion on the link between process representation and differences in simulated response (see also response to reviewer 1).

*The results discussions on the maps of temperature, don't provide analysis of map MMM (e.g. Figs 1 to 6). What are those maps? Mean of all modeling systems?*

Changes to manuscript: We will explicitly add the meaning of MMM (Multi-Model Mean, i.e. mean over all RCMs).

*Precipitation maps should be better analyzed in order to improve the conclusions on the temperature fields. Surface changes are likely to change the precipitation fields and as a feedback, the precipitation distribution is likely to have an impact the balance of radiation (downward and upward) and surface conditions, such as soil moisture and temperature. Furthermore, maps of precipitation could help on the interpretation of the discrepancies among models because this variable controls the vegetation transpiration, downward and upward short wave radiation, among others. Forest versus grass simulations for the Amazon, for instance, shows a strong change on the precipitation distribution caused by the transpiration (e.g. Ramos-da-Silva et al., J. Climate2008).*

Response: Maps of precipitation changes are already provided (fig SI), but are indeed not central in our paper due to the focus on temperature and energy balance changes. Given that precipitation changes are small for all models with no systematic pattern we can exclude the hypothesis that the spread in temperature response across models is linked to precipitation feedbacks. It is certainly thinkable that such precipitation feedbacks would be more important in a tropical context as noted by the reviewer.

Changes to manuscript: We will keep the precipitation change figure in the SI, but make a more explicit link in the paper in particular explaining that we did not find indication of strong precipitation feedbacks that could explain the discrepancies in simulated temperature response. Nevertheless for some specific models (i.e. WRF) some differences in precipitation response in different configurations can be seen and will be discussed (see also response to reviewer 1).

*The authors should provide some insights on: how the forestation affects the major synoptic systems that move across Europe? Are these atmospheric systems enhanced or weakened?*

Response: We agree that this aspect would warrant further analysis but we believe this is well beyond the scope of this paper which focuses on the mean seasonal climate response to forestation (we use monthly mean outputs in this study). Addressing changes in weather systems would require analysis at much higher temporal resolution which is the scope of an additional study currently being prepared by the LUCAS team (Strandberg et al., in prep.)

Changes to manuscript: None

*To improve the results analysis and discussion, known LSM model bias from previous studies could help on the results interpretation (e.g. Chen etal., JGR 2014).*

Changes to manuscript: Agreed, we will add a paragraph describing evaluation results from previous studies and the possible implications for our results (see also response to reviewer 1).

*Some figures should be improved. Better legends could help the readers to quickly understand the presented images. For instance, what is MMM on the maps? Furthermore, in some figures, the fonts needs to be higher to permit a better reading (e.g. Figures 7, 8, 9 and 10). Figure 7 should have a higher threshold for net radiation. It is not clear the maximum on some cases.*

Changes to manuscript: These suggestions will be implemented and the meaning of MMM (Multi-Model Mean) will be explicitly spelled out

*Further minor text corrections: Table 01 – Lateral boundary in the last column should be exponential (not expotential)*

Changes to manuscript: will be corrected

*Discussion – line 235-236 should be evapotranspiration (not evaporation)*

Changes to manuscript: will be corrected

References:

Dai, A., Giorgi, F., & Trenberth, K. E. (1999). Observed and model-simulated diurnal cycles of precipitation over the contiguous United States. Journal of Geophysical Research: Atmospheres, 104, 6377–6402. https://doi.org/10.1029/98JD02720

Gallus, W. A. (1999). Eta simulations of three extreme precipitation events: Sensitivity to resolution and convective parameterization. Weather and Forecasting, 14, 405–426.

Hu, X.-M., Xue, M., McPherson, R. A., Martin, E., Rosendahl, D. H., & Qiao, L. (2018). Precipitation dynamical downscaling over the Great Plains.Journal of Advances in Modeling Earth Systems, 10, 421–447. https://doi.org/ 10.1002/2017MS001154

Liang, X. Z., Li, L., Dai, A., & Kunkel, K. E. (2004). Regional climate model simulation of summer precipitation diurnal cycle over the United States. Geophysical Research Letters, 31, L24208. https://doi.org/10.1029/2004GL021054

Meier, R., and Coauthors, 2018: Evaluating and improving the Community Land Model's sensitivity to land cover. 400Biogeosciences, 15, 4731–4757, doi:10.5194/bg-15-4731-2018. https://www.biogeosciences.net/15/4731/2018/

---

## Author Response (AR1)

We wish to thank the anonymous reviewers for their comprehensive and constructive comments. Below, we provide detailed responses and describe the corresponding changes in the manuscript.

*Referee #1*

*The study by Davin et al. seeks to understand the impact of vegetation cover across Europe, using multiple terrestrial biosphere models coupled with regional atmospheric models. To test the sensitivity of the coupled models, they carried out theoretical simulations using minimum (i.e. grassland) and maximum forest cover in the domain. The authors found that, except for a few consistent changes, such as increase in albedo in the grassland simulations, the response of the models diverged considerably, and seemed to be more related to differences amongst the terrestrial biosphere models than the atmospheric model. This study provides an interesting insight of the drivers of divergent responses of coupled models to the same scenarios, and has a potential of becoming an important contribution to our understanding of biosphere-atmosphere interactions at regional scales. However, I think the manuscript needs some improvement and further development in the analysis and the interpretation of the results before it can be published.*

*First, the current analysis is a bit thin in assessing how atmospheric changes caused by the land cover scenarios are driving the near-surface variability across models. The authors briefly discuss these effects as potentially important, but this could be much more developed and quantified, considering that the authors are using coupled models. Just to cite one example, in the fraction of unexplained variance analysis (Fig. 10), the authors found that albedo and evaporative fraction show little explanatory power during the winter in Scandinavia. I think it should be straightforward to include some atmospheric variables such as cloud cover or incoming radiation, rainfall, or precipitable water as additional predictors. This analysis could help to quantify how surface and indirect atmospheric processes modulate surface temperature, and how this varies across regions and season.*

Response: Thank you for this suggestion. It is correct that our findings suggest that atmospheric feedbacks are more important during winter and we agree that extending our correlation analysis to atmospheric variables could help demonstrating this point more directly. Note also that this study is meant to present a first overview of the LUCAS phase 1 results focusing on temperature and energy balance at seasonal timescales (as stated in the introduction) and will be complemented by further studies analyzing more specific aspects (in particular one study investigating the role of atmospheric processes in more details is planned).

Changes to manuscript: We added incoming radiation in our correlation analysis (Fig. 12; we also switched to using 2-meter temperature in this analysis for consistency with the rest of the paper). This shows that this variable (capturing in particular possible cloud feedbacks) is able to explain some additional variance including in winter although an important part of the variance still remains unexplained in winter (which also shows the limit of this kind of simple linear regression analysis). Also we moved the shortwave radiation figures into the main text to support the discussion of radiation changes.

*In addition, as the authors noted in the text (Line 96), some combinations of models differ by only a few specific parametrisations. I think the authors could provide more insights on how the model settings and parameters are driving the differences in simulations. For example, between WRFa-NoahMP and WRFb-NoahMP simulations, the only relevant differences are the spin-up period and the sub-grid convection scheme, yet the results for precipitation in the summer are quite different (e.g Fig. 12). In*

*this case, a brief discussion on how Grell and Freitas (2014) and Kain and Fritsch (1990) differ and how the changes in surface could impact the precipitation response given the assumptions of these schemes would be very informative. Likewise, the authors included CCLM simulations with three different land surface models and a similar discussion could be included.*

Response: The main advantage of presenting results from both WRFa-NoahMP and WRFb-NoahMP is that we can clearly attribute differences in simulated response to atmospheric processes (because the same LSM is used in both configurations). However, it is really challenging to attribute differences to specific atmospheric processes because the atmospheric schemes differ in several aspects (most notably convection and microphysics). As an example, the Kain-Fritsch scheme uses a mass flux approach which rearranges each column's mass in order to remove at least 90% of the convective available potential energy (CAPE). KF is highly influenced by the boundary layer forcing, particularly surface convergence (Gallus, 1999 and Liang et al., 2004b). The Grell-Freitas scheme is based on a scale-aware method with a large-scale instability tendency closure and is more sensitive to large-scale vertical motion than the KF scheme (Dai et al., 1999). Overall, the GF scheme is known to be usually drier than the KF scheme (Hu et al., 2018). In this light, it is well possible that the slight summer precipitation decrease in WRFa-NoahMP is related to the use of the GF scheme (while precipitation is less clearly affected in WRFb-NoahMP and WRFb-CLM3.5 which use the KF scheme). It is however not possible to do more than just speculating without dedicated sensitivity experiments with WRF which would be beyond the scope of this study. Concerning the simulations based on CCLM, a recent study has examined the sensitivity to land cover changes in CLM4.5 (Meier et al., 2018) and can indeed provide insights on the interpretation of the differences seen between CCLM-CLM4.5 and the other CCLM configurations (see also response below).

Changes to manuscript: We added a discussion of the differences between WRFa-NoahMP and WRFb-NoahMP in section 3.3 focusing in particular on the role of the convection scheme for the precipitation response (these two configurations exhibit a diverging precipitation response in summer). We removed the old figure 12 so that we now discuss precipitation results only based on the difference maps in section 3.3. Indeed given that precipitation changes are small and arguably not necessarily significant, it does not seem justified to apply the cluster analysis on precipitation, which we now only apply on temperature.

*Finally, the authors need to be more careful about the role of scale when discussing the impacts of changes in land cover on the coupled biosphere-atmosphere system. For example, the authors stated that simulated changes in diurnal cycle due to forest/grass cover are the opposite of what previous observations suggest (near line 230). While I agree that this deserves further analysis, we cannot ignore that the impacts of replacing the vegetation of an entire continent on the diurnal cycle of temperature ought to be completely different from the impact of patchy deforestation/afforestation. At least in the tropics, the extent of deforestation in tropical forests may completely change the impact of land cover change in precipitation (e.g. Spracklen et al. 2018, doi:10.1146/annurev-environ-102017-030136, and references therein), and I would imagine that scale would also matter in higher latitudes and in other variables.*

Response: We totally agree that the issue of scale might be an important aspect here. Observational methods capture mainly local changes in surface energy balance and temperature due to land cover and are unlikely to reflect the type of larger scale atmospheric feedbacks that can be triggered in coupled models (especially given the drastic land cover change imposed). In other words, the apparent discrepancy between models and observations may in part arise from the fact that models and observations differ in the scale of processes considered (and we should therefore be more careful not to

attribute this discrepancy only to a lack of model realism). We agree that it is important to emphasize this point more clearly in the paper.

Changes to manuscript: The discussion on observation-based evidence has been revised and now explicitly mentions that: "… this apparent contradiction may not be only attributable to model deficiencies and could be in part related to discrepancies in the scale of processes considered in models and observations. Indeed, observation-based estimates capture mainly local changes in surface energy balance and temperature due to land cover and are unlikely to reflect the type of large scale atmospheric feedbacks triggered in coupled climate models (especially given the large scale nature of the forest expansion considered in our experiments)." We also still emphasize the need for dedicated benchmarking efforts to tackle the issue of models to observations comparison.

*Specific and minor comments*
*Abstract. It would be helpful if the authors quantified their statements. For example, when they say that the albedo decreased with forestation, they could provide the range so readers would be able to judge whether the changes are important or not. This also applies to the range of temperature responses.*

Changes to manuscript: agreed, ranges of values for temperature were added to the abstract

*Discussion. I found somewhat hard to judge most of the results because little information is provided about how these different models usually perform across Europe. For instance, it would be very helpful to know whether some of them always underestimate rainfall or overestimate evapotranspiration, for example. Ideally the authors could show some model assessment with known benchmarks, but I understand that the model cannot be validated for idealised simulations. One suggestion would be to include one or two paragraphs describing previous studies using these models.*

Response: As noted by the reviewer, the type of extreme land cover experiments performed here are not suitable for classical evaluation purpose. However, most of the RCMs used in our study have been part of EURO-CORDEX and have been evaluated in this context (e.g. Kotlarski et al 2014; Davin et al. 2016). Although for a given RCM the version and configuration used in our study may differ from the published EURO-CORDEX counterpart, the systematic biases that have been previously identified are still relevant here (e.g. predominant cold and wet biases for most European regions with the exception of Southern Europe in summer where the opposite occurs). We agree that this context should be provided.

Changes to manuscript: We added a paragraph in section 2.1 describing previous evaluation results and the type of systematic biases found in these RCMs.

*Line 36. Give concrete examples of atmospheric processes that dictate more directly the simulated response.*

Changes to manuscript: We changed this sentence now explicitly referring to radiation/cloud feedbacks building upon the new correlation analysis introduced in this version.

*Line 51. A better justification is needed for the opening sentence. One could still see some regional effects of land cover and land use change in global models, especially the extreme land cover scenarios used in this study.*

Response: It is true that global models can also represent some regional LUC effects as we indeed summarize in the first paragraph. We are certainly not arguing that LUC should be included in RCMs rather than in GCMs, obviously it should be considered in both types of models. But the current situation is that RCMs intercomparison projects typically ignore LUC effects. We are therefore arguing that this should be remedied, one of the added values being (beside improving the consistency in forcings included in global and regional experiments) that new insights could be gained given the higher resolution at which RCMs operate (e.g. enabling to capture such local to regional effects in more details). While one could argue that this resolution issue is not critical in the extreme scenarios analyzed in the phase 1 of LUCAS, this nevertheless provides an essential motivation for the LUCAS project in general and beyond phase 1.

Changes to manuscript: We changed the opening sentence to avoid giving the impression that regional effects can only be seen in RCMs: "In this light, it is particularly important to represent LUC forcings not only in global climate models but also in regional climate simulations."

*Line 110-112. This is fine for the experiment, but the current approach does not completely prevent placing trees in areas naturally dominated by grasses or shrubs like some of the Mediterranean maquis.*

Response: It is certainly true that the FOREST map does not represent a potential vegetation map. Our intention was indeed to generate a map of maximum forest coverage, which we thought was more meaningful to explore the full forestation potential. There are already many examples of places in Europe where trees are growing where they would not be the naturally occurring vegetation type simply because of human intervention (assisted afforestation, forest management, fire suppression, etc). Our forest map is therefore less conservative in terms of potential for tree expansion than a potential vegetation map, which is in line with the idea of considering both reforestation and afforestation potential (note that we still exclude forest expansion over drylands that would likely imply drastic irrigation measures).

Changes to manuscript: We added a paragraph in section 2.2 to clarify this point.

*Section 3.1. Figures 3-6 are interesting, but I think you could move some of them to the SI, and bring some of the atmospheric figures that could help explaining these differences to the main text (e.g. net radiation or precipitation).*

Changes to manuscript: We moved the shortwave radiation figures into the main text, but we chose to keep 3-6 as they are described in the text and the total number of figures (now 13) is not unreasonably high.

*Line 162. One interesting feature is that all CLM runs (CCLM, WRF, RegCM) seem to show increases in ET during the spring, but not during the summer. Could this be an extreme response to drought stress, like the beta factor being too low that stomata are closed for most of the summer?*

Response: This behavior has been analyzed in details in an offline study with CLM4.5 (Meier et al., 2018) which indeed concluded that there is a too strong water limitation effect (too low beta factor in forest compared to open land) in summer in CLM4.5, while ET is higher in spring under forested conditions. The fact that this occurs also in the context of offline simulations confirm that this is an intrinsic feature of the CLM land surface model. Meier et al., 2018 tested various modifications to alleviate this issue but these modifications were not included in the coupled RCMs which are based on the default version of CLM.

Changes to manuscript: We included a discussion on this particular behavior in CLM4.5 in the discussion section.

*Lines 165-170. Add references to figures/tables.*

Changes to manuscript: references to figures were added

*Lines 169-170. The last sentence belong to discussion instead of results, and I think this deserves to be expanded a bit more: this is likely to be related to my previous comment. One possibility could be that trees may be transpiring too much during the spring then running out of water during the summer. However, this is not so clear over land in Fig. S11. The strongest negative tendencies seem to be over the Mediterranean Sea and Black Sea. Just to confirm, do the averages in Figures 7–10 exclude grid cells over water?*

Changes to manuscript: This sentence has been moved to the discussion and expended. In particular the discussion is complemented by results from Meier et al., 2018 (see above). Averaged values indeed include only land points as mentioned in the first sentence of section 3.2.

*Lines 185-189; lines 195-196; 212-213. If most of the variance in winter is not explained by albedo and evaporative fraction, then what is causing the variability? Could the variations be attributed to changes in weather patterns? The authors could and should include predictors that were not so surface-centric. The authors suggest that precipitation has no consensus amongst models (unsurprising, this is often more uncertain than other meteorological variables).*

Response: It would be beyond the scope of this manuscript to analyze weather patterns given the focus on the mean seasonal response in temperature and surface energy balance (weather phenomena would require outputs at much higher temporal resolution). But we will expand the discussion on the role of atmospheric processes as already discussed above.

Changes to manuscript: Incoming radiation was added to the analysis of variance (see above). We also added a paragraph describing the role of precipitation changes in section 3.3.

*Lines 228-232: I do not necessarily see an inconsistency between model and independent observations. Europe is not a grassland continent nor a forest continent, and when an entire continent changes land cover, then the response will be very different from patches of forest next to patches of grassland that are side by side.*

Response: We agree that the scale effect can play an important role here.

Changes to manuscript: This paragraph has been reformulated (see response above).

*Line 242-243. I agree with this sentence, but the authors did not provide any insight on this. The authors could at least use the simulations in which differences are contained to indicate some of the origin of uncertainties. This would make the discussion much more interesting and informative, and go beyond the "models showed little or no agreement" storyline that we often see in model intercomparison papers.*

Response: This sentence is about differences between surface versus 2-meter temperature. We agree that we did not give a lot of insights on this point (other than showing maps of surface temperature changes in the SI), but a separate study within LUCAS is exploring specifically the issue of surface versus 2-meter temperature (Breil et al., submitted). We therefore decided to focus only on 2-meter temperature results here and consistently use only 2-meter temperature across the entire manuscript.

Changes to manuscript: This sentence was removed as well as the SI figure for surface temperature.

*Table 1.*
*• I found this table a bit hard to read, I wonder if it would make it easier to separate the atmospheric and land surface settings.*
*• Was there any reason why the spin-up period was different for the three WRF simulations?*
*• In the greenhouse gas row, What is the difference between historical and constant? I assume historical means time varying, but this should be clarified, and a citation should be provided. For the simulations with constant values, provide these values, at least for CO2.*

Response: The spin-up in WRFb-Noah and WRFb-CLM3.5 is shorter only due to computational constraints.

Changes to manuscript: We now included a separation within the table between "land settings" and "atmospheric settings". We condensed the table using acronyms for the definition of vegetation types which overall improved the readability. We provide the constant CO2 values and the reference for the transient GHG.

*Figures 1-6. What is MMM? I assume that it is the average response. Explain this in the figure captions.*

Changes to manuscript: We added in section 2.1 that MMM means Multi-Model Mean.

*Figure 9. Consider adding a similar panel for the Mediterranean.*

Changes to manuscript: Thank you for this suggestion. We added a second panel for the same region (Scandinavia) but illustrating the relation between changes in albedo and temperature since this is one of the important driving factor described in the text.

*Referee #2*

*The paper addresses relevant scientific questions. There was a strong effort of several groups in order to provide modeling results and evaluate the modeling physics of several systems. The intercomparison of the surface and atmospheric modeling systems and the two experiments (i.e. total FOREST x GRASS) provides an interesting tool for modeling improvement, mainly on the understanding the LSM impacts and feed-backs. The modeling results provide a new original contribution. The methodology is appropriated for the goals of the project. It provides variable Land Surface Systems and various atmospheric options. The paper structure has a good flow and fluency. See some questions & suggestions below.*

*The results show that there were several differences among the modeling systems used on the inter-comparison. I think on the methodology more information should be provided by the authors, mainly on the soil types and vegetation parameters such as the maximum/minimum stomatal resistance,*

*vegetation height, and roots depth. Those parameters could allow improvements on the major conclusions. Furthermore, it could help the reproduction of the numerical experiments.*

Response: It is certainly conceivable that vegetation parameters could explain some of the differences between models. However, we don't think that listing these parameters will significantly shed light on the potential sources of model differences (mainly because it will not be possible to disentangle the effect of these parameters from the effect of other parameters/parameterization). That said, one particular feature of our RCM ensemble is that some RCMs are used in different configurations with a limited set of differences (e.g. WRF family) which can be an advantage when trying to attribute differences in model response to specific processes. We will therefore strengthen the discussion around specific parameterization differences in these models in order to make a clearer link between process representation and differences in simulated response (see also response to reviewer 1). Concerning specifically the role of vegetation parameters, a recent study using CLM4.5, which is also use here in two RCMs, investigated the role of vegetation/root parameters and water uptake parameterization on the simulated effect grass to forest conversion (Meier et al., 2018). Since these results can help understand the behavior of the RCMs using CLM we will provide a discussion with a link to this study.

Changes to manuscript: In various places we added more discussion on the link between process representation and differences in simulated response. We added for instance a paragraph on the role of the convection scheme in WRF (section 3.3) and a paragraph on the role of specific land parameters (root distribution and photosynthetic parameters) in the discussion section (see also response to reviewer 1).

*The results discussions on the maps of temperature, don't provide analysis of map MMM (e.g. Figs 1 to 6). What are those maps? Mean of all modeling systems?*

Changes to manuscript: We explicitly added the meaning of MMM (Multi-Model Mean, i.e. mean over all RCMs) in section 2.1.

*Precipitation maps should be better analyzed in order to improve the conclusions on the temperature fields. Surface changes are likely to change the precipitation fields and as a feedback, the precipitation distribution is likely to have an impact the balance of radiation (downward and upward) and surface conditions, such as soil moisture and temperature. Furthermore, maps of precipitation could help on the interpretation of the discrepancies among models because this variable controls the vegetation transpiration, downward and upward short wave radiation, among others. Forest versus grass simulations for the Amazon, for instance, shows a strong change on the precipitation distribution caused by the transpiration (e.g. Ramos-da-Silva et al., J. Climate2008).*

Response: Maps of precipitation changes are already provided (fig S5), but are indeed not central in our paper due to the focus on temperature and energy balance changes. We nevertheless added a paragraph to describe precipitation change and potential feedbacks.

Changes to manuscript: We added a paragraph on precipitation and precipitation feedbacks in section 3.3 mentioning that precipitation changes are small but may play a role in the context of specific models such as WRF: "We note however that precipitation changes are small in all RCMs with no clear consensus among models (Fig. S5). One possible exception is the summer precipitation decrease in WRFa-NoahMP which could be related to the use of the Grell-Freitas convection scheme (Table 1),

while precipitation is less affected in WRFb-NoahMP and WRFb-CLM3.5 which use the Kain-Fritsch scheme. The stronger summer temperature increase in WRFa-NoahMP compared to WRFb-NoahMP and WRFb-CLM3.5 may therefore be linked to this precipitation feedback."

*The authors should provide some insights on: how the forestation affects the major synoptic systems that move across Europe? Are these atmospheric systems enhanced or weakened?*

Response: We agree that this aspect would warrant further analysis but we believe this is well beyond the scope of this paper which focuses on the mean seasonal climate response to forestation (we use monthly mean outputs in this study). Addressing changes in weather systems would require analysis at much higher temporal resolution which is the scope of an additional study currently being prepared by the LUCAS team (Strandberg et al., in prep.)

Changes to manuscript: Although we did not address this point directly, we added incoming shortwave radiation in the correlation analysis thus proving additional insights on atmospheric processes (see response to reviewer #1).

*To improve the results analysis and discussion, known LSM model bias from previous studies could help on the results interpretation (e.g. Chen etal., JGR 2014).*

Changes to manuscript: Agreed, we added a paragraph in section 2.1 describing previous evaluation results and the typical systematic biases present in RCMs which provides an important context for this study (see also response to reviewer 1).

*Some figures should be improved. Better legends could help the readers to quickly understand the presented images. For instance, what is MMM on the maps? Furthermore, in some figures, the fonts needs to be higher to permit a better reading (e.g. Figures 7, 8, 9 and 10). Figure 7 should have a higher threshold for net radiation. It is not clear the maximum on some cases.*

Changes to manuscript: We clarified the meaning of MMM (Multi-Model Mean) and we changed the range of figures.

*Further minor text corrections: Table 01 – Lateral boundary in the last column should be exponential (not expotential)*

Changes to manuscript: corrected

*Discussion – line 235-236 should be evapotranspiration (not evaporation)*

Changes to manuscript: corrected

Additional notes from the authors:

1) A minor correction was made to the REMO-iMOVE simulations. The new version of these simulations were integrated to this revised version resulting only in minor differences without any changes to the conclusions.
2) In the submitted version, the wrong figure was used for panel b in figure 13 (due to a mistake in the season considered). This is corrected in this revised version.

[revised manuscript text omitted]

---

## Author Response (AR2)

Dear Editor,

We have corrected the issues with the figures as indicated below.

"Dear Authors, I have received the reports from the reviewers. I am happy to say that they are satisfied with your revisions. I think this will be an important addition to the literature. I have 2 minor points: 1. The legend fonts of the Figures 9, 10, 11, 12 are too small to be legible. Also the text labels in fig 11 are overlapping. "

*Fonts have been increased in all those figures and the text labels in fig 11 are not overlapping anymore.*

"2. The SWin radiation bars in Figure 12 are impossible to see because their color is too similar to the alb+EF+SWin bar in the background."

*The color has been changed and is now more distinguishable from the background color.*